# Unraveling radiation resistance strategies in two bacterial strains from the high background radiation area of Chavara-Neendakara: A comprehensive whole genome analysis

**Sowptika Pal[1], Ramani Yuvaraj[2⊙], Hari Krishnan[2⊙], Balasubramanian Venkatraman[2], Jayanthi Abraham[3], Anilkumar Gopinathan[1] \***

1 Molecular Endocrinology Laboratory, School of Biosciences and Technology, Vellore Institute of Technology, Vellore, Tamil Nadu, India, 2 Radiological and Environmental Safety Division, Indira Gandhi Centre for Atomic Research, Kalpakkam, Tamil Nadu, India, 3 Microbial Biotechnology Laboratory, School of Biosciences and Technology, Vellore Institute of Technology, Vellore, Tamil Nadu, India

⊙ These authors contributed equally to this work.
\* ganilkumar@vit.ac.in

## Abstract

This paper reports the results of gamma irradiation experiments and whole genome sequencing (WGS) performed on vegetative cells of two radiation resistant bacterial strains, *Metabacillus halosaccharovorans* (VITHBRA001) and *Bacillus paralicheniformis* (VITH-BRA024) ($D_{10}$ values 2.32 kGy and 1.42 kGy, respectively), inhabiting the top-ranking high background radiation area (HBRA) of Chavara-Neendakara placer deposit (Kerala, India). The present investigation has been carried out in the context that information on strategies of bacteria having mid-range resistance for gamma radiation is inadequate. WGS, annotation, COG and KEGG analyses and manual curation of genes helped us address the possible pathways involved in the major domains of radiation resistance, involving recombination repair, base excision repair, nucleotide excision repair and mismatch repair, and the antioxidant genes, which the candidate could activate to survive under ionizing radiation. Additionally, with the help of these data, we could compare the candidate strains with that of the extremely radiation resistant model bacterium *Deinococccus radiodurans*, so as to find the commonalities existing in their strategies of resistance on the one hand, and also the rationale behind the difference in $D_{10}$, on the other. Genomic analysis of VITHBRA001 and VITHBRA024 has further helped us ascertain the difference in capability of radiation resistance between the two strains. Significantly, the genes such as *uvsE* (NER), *frnE* (protein protection), *ppk1* and *ppx* (non-enzymatic metabolite production) and those for carotenoid biosynthesis, are endogenous to VITHBRA001, but absent in VITHBRA024, which could explain the former's better radiation resistance. Further, this is the first-time study performed on any bacterial population inhabiting an HBRA. This study also brings forward the two species whose radiation resistance has not been reported thus far, and add to the knowledge

**Data Availability Statement:** All relevant data are within the manuscript and its Supporting Information files.

**Funding:** Authors AG and SP received funding from the Ministry of Earth Sciences (MoES), Government of India as per the order Moes/36/OOIS/extra/26/2013. URL https://www.moes.gov.in/ This is to state that the funder has no role in the study design, data collection and analysis, decision to publish, or preparation of the manuscript.

**Competing interests:** The authors have declared that no competing interests exist.

on radiation resistant capabilities of the phylum Firmicutes which are abundantly observed in extreme environment.

## Introduction

Ionizing particles like X-rays and gamma rays are known to directly interact with DNA of the target organism, leading to multiple types of damages, thereby causing deleterious effects, including death. Chromosomal DNA lesion such as DNA single strand breaks (SSB), double strand breaks (DSB) and damaged bases are considered to be hazardous effects of irradiation [1]. DSBs, frequent enough, could result in chromosomal aberrations, leading to genome instability and cell death, unless appropriately repaired [2]. Another aftermath of irradiation is the production of reactive oxygen species (ROS) which could damage all the biomolecular components in the cell, resulting in protein malfunction (through protein oxidation and amino acid carbonylation) [3].

To counter the hazardous impact of exposure to ionizing radiation, microorganisms, irrespective of their habitats and phylogenetic status, have adopted defensive mechanisms. Extremophilic bacteria are known to survive intense doses of radiation (~15 kGy) that is lethal to most organisms, and their radiation resistance capabilities are about 1000 and 3000 times greater than that of typical eukaryotes and humans, respectively [4]. Reports are also available on those bacteria having low levels of tolerance (<1 kGy); *Escherichia coli*, for instance, displaying a $D_{10}$ value, as low as 0.7 kGy and *Shewanella oneidensis* with a low $D_{10}$ value of 0.07 kGy [5] were investigated in detail, using high-throughput techniques [6, 7]. These studies have afforded us valuable information, on the cellular mechanisms adopted by these microbes to cope with adverse radiation pressures.

Earlier researchers have attempted to identify the key determinants of radiation resistance among bacteria, at physiological, cellular and biochemical levels, using the highly resistant *Deinococcus radiodurans*, as the model organism. Here, nutrient rich conditions have been suggested to be crucial for the irradiated cells to revive, thus becoming a physiological determinant [8]. Again, in *D. radiodurans*, redundancy of genetic information in conjunction with its efficient recombination system, resulted from polyploidy is suggested to act as the cytological determinant [9]. Further, the central role played by the down-regulation of isocitrate-fumarate steps in tricarboxylic acid (TCA) cycle for the recovery from ionizing radiation [5] and the predominance of enzymatic/non-enzymatic metabolites comprising manganese complexes for quenching ROS [10] are seen to be the determinants from metabolic perspectives. Another factor that is suggested to act as a determinant is the protection against radiation–induced protein oxidation, as demonstrated in *D. radiodurans* [11]. Results of all the aforementioned studies put together, it would only be reasonable to believe that radiation tolerance could be resulted from a combination of different mechanisms.

However, notwithstanding the wealth of information on the highly radiation resistant bacteria such as *Deinococcus radiodurans*, *Deinococcus deserti*, *Keinococcus radiotolerans*, *Rubrobacter radiotolerans* [12–15] and some of the low-level radiation tolerant ones such as *S. oneidensis* and *E. coli*, there are only very few attempts with respect to the adaptive strategies of moderately radiation resistant bacteria (MRRB) [that can tolerate 1 to 9 kGy]. Attempts that have been made on these bacteria [16–22], were restricted to their isolation and identification from varied habitats; the exact mechanism of resistance among MRRB still remains to be a knowledge gap. Obviously, this missing information is very much pertinent for a better

assessment of not only the impact of radiation, but also to evolve remedial measures to counter the deleterious effects of radiation exposure. Further, due to the stress tolerance capacity of medium resistant bacteria it could be used for bioremediation of radionuclide from nuclear waste, contaminated soil and mines and they could be recycled as raw material, thus creating a 'circular economy' [23]. Several of the extremophiles are known to produce low-molecular weight substances (extremolytes) and proteins (extremozymes) as their secondary metabolites which are of high relevance for white, grey, and red biotechnological sectors [24, 25]. These qualities of radiation resistant extremophiles have become 'researcher hotspot' of late; and there have been strong suggestions that capacity to produce potentially useful secondary metabolites is not restricted to highly resistant varieties (such as *Deinococcus* sp.), but can occur among microbial taxa of medium resistance [25].

Considerable degrees of similarities as well as diversities exist among microbes in their ways to counter the ill-effects of radiation. *D. radiodurans*, for instance, relies on *ddrA*, *ddrB*, *ddrO* and *irrE* (also known as *pprI*) genes which are exclusive to them for efficient DNA repair [26]. Further, majority of the members of the genus *Deinococcus* (*D. radiodurans*, *D. geothermalis*, *D. deserti* and *D. murrayi*) and other bacteria including *Rubrobacter radiotolerans* [14] and *Micrococcus luteus* [16] depend highly on RecFOR system for recombinational repair of DNA while others such as *Keinococcus radiotolerans* [27], *Rhodococcus erythropolis* [28] and *Promicromonospora panici* [17] show the presence of both RecFOR and RecBCD pathways. Yet another variety, *Methylobacterium radiodurans*, depends only on RecBCD pathway [22] for repair mechanisms. For optimized maintenance of Mn/Fe ratio, which is crucial for ROS removal to ensure resistance, radio-resistant bacteria such as *D. radiodurans*, *D. geothermalis* [29] and *P. panici* [17] possess both MntABCD and NRAMP manganese transporter system. *R. radiotolerans* [14], *D. peraridilitoris* and *D. puniceus* [29], on the other hand, rely only on MntABCD transporter system. Such diversities among microbes in their mechanisms for radiation tolerance, make it difficult to draw a generalized picture on their adaptive strategies. This situation makes it imperative to have more investigations on the defence mechanisms of varied kinds of microbes (bacteria, for instance), representing diverse phylogenetic status and levels of resisting capabilities. Keeping this in view, a gamma irradiation experiment, followed by whole genome sequence (WGS) studies was performed on two bacterial strains (VITHBRA001 and VITHBRA024) as an attempt to assess the genes and their possible pathways that could afford them efficient radiation protection involving DNA repair (including homologous recombination, nucleotide excision repair, base excision repair, mismatch repair), optimized Mn/Fe ratio and enzymatic and non-enzymatic anti-oxidant systems to withstand against medium level radiation exposure. Most of the organisms subjected to radiation tolerance studies have been chosen from various extremophilic habitats such as snow-capped mountains [30], deserts [31] and hot spring [14] but none of them from natural background radiation area, making this a first-time report of its kind.

## Materials and methods

### Preparation of vegetative cell for irradiation

Out of 35 species identified from HBRA and reported earlier from our laboratory [32], 32 of them were exposed to acute dose of ionizing (Gamma) radiation, in their vegetative phase, using cobalt-60 ($^{60}$Co), at the radiation facility of Indira Gandhi Centre for Atomic Research (IGCAR), Kalpakkam. The cultures were grown in 100 ml of Zobell Marine broth 2216 (ZMB) at 35°C, kept in a shaker at 120 rpm to attain exponential growth phase of ~1.0 OD. 2 ml aliquots of the culture were collected for heat treatment and control, centrifuged at 3500 rpm and the pellets were washed once with phosphate buffer saline (PBS), subsequently

resuspended in PBS and heat treatment was given at 80˚C for 15 min; control was not exposed to heat treatment. The aliquots were then serially diluted to $10^{-6}$ dilution and each dilution was plated with 20 μl spot on Zobell marine agar (ZMA) media in triplicates. The plates were then incubated for 24 hours at 35˚C. With this method, the vegetative cells would die and any spore formed by the bacteria would survive and grow on the plate [33]. The absence of any growth on the media plate due to heat treatment confirms that the cultures were in vegetative phase.

## Exposure to ionizing radiation and $D_{10}$ value calculation

The irradiation was carried out on triplicates of samples, in an ascending order of 1 kGy each, to a maximum of 5 kGy. For, irradiation, the cultures were grown in 100 ml of ZMB at 35˚C, kept in a shaker at 120 rpm to attain exponential growth phase of ~1.0 OD and collected in 2 ml vials (in triplicate) for each dose and control (no irradiation). The 2 ml vials were centrifuged at 3500 rpm for 15 mins, after which the supernatant was discarded. Fresh 2 ml ZMB was added to the vials and the cell pellets were resuspended with gentle tapping to cause minimal harm to the cells. The collected samples were then maintained in ice at 4˚C until irradiated. The entire experiment was conducted in ice to reduce heating and any chances of metabolic activity. The samples were irradiated at 0.77 kGy/hr dose rate and collected at an interval of every one kGy; 1 kGy took ~1 hr 18 min, 2 kGy took ~2 hr 36 min, 3 kGy took ~3hr 54 min, 4 kGy took ~5 hr 12 min and 5 kGy took ~6 hr 30 min. Similar dose rate of 0.7 kGy/hr was used to irradiate *Bacillus* samples [34]. Care was taken to maintain accuracy in dosage measurements, so as to avoid chances of experimental error. The survival of the irradiated sample was observed by first serial diluting each vial to $10^{-6}$, and then plating 20 μl of each dilution on ZMA media by Miles and Misra (1938) method. The plates were incubated at 35˚C overnight and visible colonies were counted. Mean values were calculated using the bacterial cells number obtained from triplicate data. Survival graph of log ($N/N_0$) was plotted against the radiation dose, wherein N represents the number of bacterial cells survived after irradiation and $N_0$ represents the number of bacterial cells without irradiation. A linear regression line was obtained, the slope of which was used to calculate the $D_{10}$ value (decimal reduction dose i.e., the radiation dose required to inactivate 90% of a viable microbial population or reduce the population by a factor of 10) [35].

## DNA extraction for WGS

A single colony from pure culture of the bacterium was inoculated in 20 ml ZMB and kept in shaker (120 rpm) at 35˚C overnight, to be used for DNA extraction using the Qiagen All-Prep® Bacterial DNA/RNA/Protein kit. The integrity and purity of DNA was checked using NanoDrop™ 2000 Spectrophotometer (ND2000, ThermoFisher Scientific), followed by agarose gel electrophoresis.

## Library preparation and whole genome sequencing

The library was prepared using NEBNext® Ultra™ II FS DNA Library Prep Kit, subsequently sequencing in Illumina HiSeq X for paired end reads; the kit comes with an enzyme mix which helps fragment the DNA (500 ng was used) in desired size (in this case 275–475 bp), repairs and produces blunt ends on the fragments and finally adds a 3′ poly-A tail. To these adenylated fragments, looped adapters were ligated using ligation master mix, and consecutively the loop was cleaved with USER (uracil-specific excision reagent) enzyme. AMPure XP beads were used to purify the fragments and finally a library of 350–600 bp was obtained. Furthermore, the DNA was amplified by 8 cycles of PCR with the addition of NEBNext Ultra II Q5 master

mix, and NEBNext® Multiplex Oligos for Illumina to facilitate multiplexing while sequencing. The library was purified again using 0.9X AMPure XP beads, and quality control was performed using Qubit 3.0 Fluorometer (Thermo Fisher Scientific) and Agilent D5000 ScreenTape System.

## Quality control, assembly and annotation

The KBase pipeline [36] for prokaryotic genome assembly was used to check the quality of the Illumina reads files and assemble the genome. Tools such as FastQC v0.11.9 [37] was used to check the quality of the raw reads, followed by BBtools v38.22, with a view to trim both the ends of the reads which had low quality bases ($< 33$ Phred score, default parameters), and also to remove any duplicate reads. The filtered reads, so obtained, were assembled using assemblers SPAdes v3.15.3 [38], Unicycler v0.4.8 [39] and MEGAHIT v1.2.9 [40], with 500 bp as minimum contig length cut-off. The assemblies were checked for its quality, with QUAST v4.4 [41] which gave us the details of the assembled genome. The maximum usage of the sequenced reads to obtain the final assembly was checked using Bowtie2 v2.3.2. The best assembled genome was selected based on parameters of N50, L50, BUSCO completeness and percentage alignment of the assembly against the reads. The assembled genome was annotated using the Prokka pipeline [42], NCBI prokaryote genome annotation pipeline (PGAP) [43] and functional annotation by RAST seed server [44].

## Identification of the genome

The final assembled genome FASTA file was uploaded in Type Strain Genome Server (TYGS) (https://tygs.dsmz.de) which efficiently identifies prokaryotic species using a type strain genomic database and accurately classifies its taxonomy using a digital DNA-DNA hybridization calculation (dDDH) [45]. The server also provided a phylogenetic tree, revealing the closeness of the genome in question with the type strains in database. The tree was represented using iTol online tool version 6.8.1 [46].

## CDS picking for analysing radiation resistance pathway

Coding sequences (CDSs) or proteins of interest were compared manually for clarity of annotation among the annotated data generated by Prokka, NCBI and RAST and the sequences used for further analysis were taken from Prokka. Additionally, KEGG analysis using BlastKoala online software [47] and COG analysis using PyPI cogclassifier version 1.0.5 (a project developed and maintained by the Python community) [48] helped identify more potential genes which were annotated as hypothetical by the annotation pipelines. Further, protein functional domains were also identified and confirmed using online NCBI conserved domain (CD) search tool [49]. Uniprot online version (release 2023_04) was used for protein multiple sequence alignment which aligns using Clustal Omega version 1.2.2 and percentage identity was also calculated in Uniprot. The protein sequence of organisms (excluding the candidate strains in this study) were downloaded from Uniprot and phylogenetic tree with 1000 bootstraps replicates were constructed using MEGA X version 10.1.8 [50]. The trees were represented using iTol online tool [46]. The pigment producing genes were identified using online antiSMASH secondary metabolite searching software version 7.0 [51]. Standalone BLAST version 2.12.0 [52] was used to find any homologous genes of interest within the protein FASTA file of the candidate strains under the present study.

### Statistical analysis

The normality of the values for survivability graph was estimated with Shapiro-Wilk normality test using R stats package. The linear regression analysis of the survivability graph using R stats package provided us with R squared ($R^2$) value, confidence intervals and significance level which helped substantiate the data statistically.

## Results and discussion

### Induced radiation experiments on VITHBRA001 and VITHBRA024, isolates from HBRA

The plate culture experiments on the irradiated samples in their vegetative phase revealed that both the strains could withstand a radiation dose up to 5 kGy. For VITHBRA001, linear regression analysis of the plotted graph for survivability has shown its $D_{10}$ value as 2.32 kGy (Fig 1A), which is 8.9 times more radiation resistant than *E. coli* (ATCC25922 with $D_{10}$ = 0.26) and 33 times more resistant than radiation sensitive bacteria *S. oneidensis* MR1 ($D_{10}$ = 0.07) [5]. Further, the other bacterial strain VITHBRA024 with $D_{10}$ value of 1.42 kGy was found to be 5.4 times more resistant than *E. coli* and 20 times more resistant than *S. oneidensis* (Fig 1B). Any bacterial strain which has a $D_{10}$ value of >1 kGy is considered as ionizing radiation resistant bacteria (IRRB) [53], making both the strains eligible to be considered as 'radiation resistant bacteria'. Relying on the proven estimates by Gerwen *et al.* (1999), the resisting capability of sporulated forms of VITHBRA001 and VITHBRA024 could be more, at least 3–5 order of magnitude than their vegetative forms [54].

### Statistical interpretation of the results

As per the statistical analysis of the survival graphs, confidence intervals (CI) were obtained as CI = -1.050068333 ± 0.650 for VITHBRA001 and CI = -2.294401667 ± 1.146 for VITHBRA024 at the significance level of 95%. Shapiro-Wilk normality test suggest that at 95% significance level, the raw data are normally distributed. Additionally, $R^2$ values for both the graphs suggest that the survival of the organisms as dose-dependent. We could also determine that the

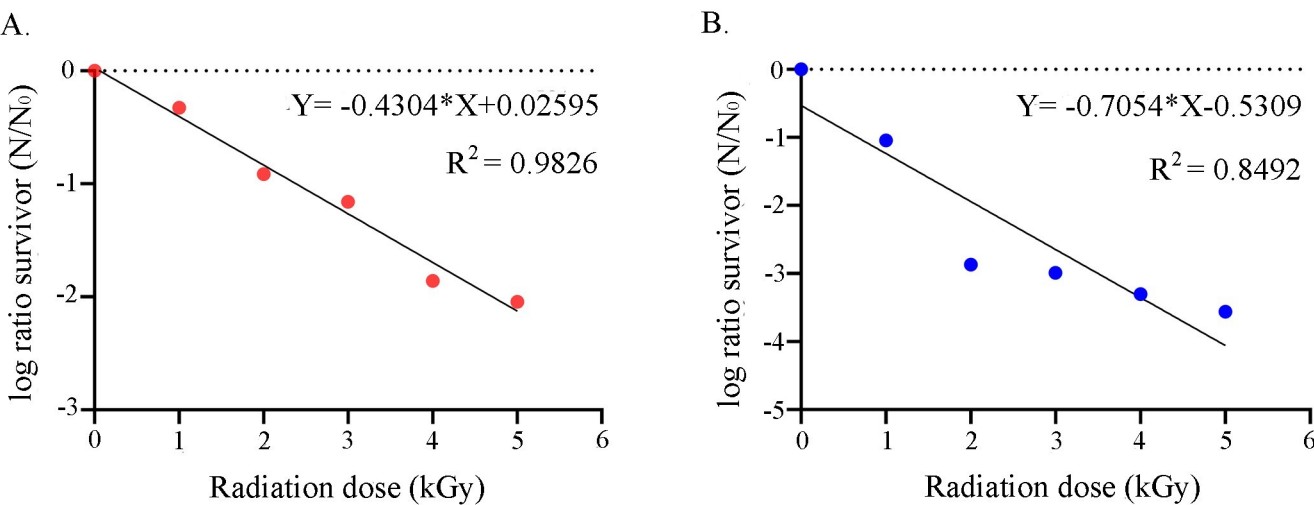

**Fig 1. Ionizing radiation survival graph of VITHBRA001 and VITHBRA024.** Figure (A) and (B) show the survival graph of VITHBRA001 and VITHBRA024 respectively wherein the regression equation of the linear trendline (solid black line) helped us calculate the $D_{10}$ value. The graph was drawn using GraphPad Prism version 8.4.2 (https://www.graphpad.com).

regression line is significantly non-zero with a confidence of $> 99\%$ suggesting that the $D_{10}$ values of both the strains are significantly different, further reiterating the difference in radiation tolerance existing between the two strains.

## WGS of VITHBRA001 and VITHBRA024

Illumina HiSeq platform sequencing of VITHBRA001 and VITHBRA024 provided us with 11,867,862 reads and 41,810,468 reads respectively with almost 97% and 91.35% of the reads passing the Phred score of >30. On passing the reads through RQCFilter pipeline of BBTools, 11,859,408 reads and 41,273,116 reads were obtained respectively, to further process them for assembly. Three assemblers were tried to obtain the best possible assembly. Up on comparison using QUAST assessment tool (Table 1) and aligning the assembled sequences with the reads using Bowtie2, MEGAHIT assembled data were shown to be the most ideal for VITHBRA001, while SPAdes assembled data with default settings was found to be the best fit for VITH-BRA024. For VITHBRA001, MEGAHIT gave an output of 52 contigs with total genome length of 5,205,284 bp and GC content of 35.9%; the largest contig size was of 432,869 bp, with an N50 of 222,429 bp and L50 of 8 contigs, and no incorporation of N's. 99.49% of the reads also mapped against the assembled genome by MEGAHIT as compared to 99.19% by SPAdes and 98.96% by Unicycler. In order to see the completeness of the genome, BUSCO showed 99.3% complete using bacteria_obd9 lineage and CheckM showed 98.57% complete using genus *Bacillus* (g_Bacillus) as the marker lineage having 101 genomes. In case of VITHBRA024, the

**Table 1. QUAST analysed assembled genome information.**

| Strains under study | VITHBRA001 | VITHBRA024 |
|---|---|---|
| Assembly by | MEGAHIT v1.2.9 | SPAdes v3.15.3 |
| # contigs ($> = 0$ bp) | 52 | 27 |
| # contigs ($> = 1000$ bp) | 48 | 20 |
| # contigs ($> = 5000$ bp) | 43 | 16 |
| # contigs ($> = 10000$ bp) | 42 | 16 |
| # contigs ($> = 25000$ bp) | 33 | 13 |
| # contigs ($> = 50000$ bp) | 26 | 11 |
| Total length ($> = 0$ bp) | 5204837 | 4316792 |
| Total length ($> = 1000$ bp) | 5201792 | 4312383 |
| Total length ($> = 5000$ bp) | 5188813 | 4306645 |
| Total length ($> = 10000$ bp) | 5181325 | 4306645 |
| Total length ($> = 25000$ bp) | 5046595 | 4249783 |
| Total length ($> = 50000$ bp) | 4771361 | 4162999 |
| # contigs | 52 | 27 |
| Largest contig | 432869 | 1429755 |
| Total length | 5204837 | 4316792 |
| GC (%) | 35.92 | 45.77 |
| N50 | 222429 | 791221 |
| N90 | 52243 | 108109 |
| L50 | 8 | 2 |
| L90 | 25 | 8 |
| # Ns per 100 kbp | 0.00 | 4.33 |

All statistics are based on contigs of size $> = 500$ bp, unless otherwise noted (e.g., "# contigs ($> = 0$ bp)" and "Total length ($> = 0$ bp)" include all contigs).

*de novo* assembly with SPAdes gave an output of 27 contigs with a total genome length of 4,316,792 bp and GC content of 45.7%, the largest contig size being 1,429,755 bp with an N50 of 791,221 bp and L50 of 2 and incorporation of 4.33 Ns per 100 kbp. A total of 98.39% of the reads also mapped against the assembled genome (VITHBRA024) followed by BUSCO completeness of 98% using bacteria_odb9 as lineage dataset and CheckM showed 99.59% completeness using genus *Bacillus* (g_Bacillus) as the marker lineage having 93 genomes.

## Identification of the strains

From our previous data on 16S rDNA identification of bacteria [32], it was observed that VITHBRA001 maintained 100% bootstrap alignment with the species *Metabacillus halosaccharovorans*. Even though 16S rDNA is considered the gold standard for identification of bacteria, whole genome sequencing is currently being used to obtain the accurate identification of bacteria. After the de novo assembly of the reads of these bacterial strains, it was submitted in the TYGS online website which gave 100% bootstrap support with *M. halosaccharovorans* type strain DSM 25387 (Fig 2A) and provided a digital DNA-DNA hybridization (dDDH) value of 81.7% with a confidence interval of 77.8–85.1% (using the Genome Blast Distance Phylogeny approach) affirming that the strain belongs to this species. DDH value >70% confirms the identification of a strain to a particular species which was earlier done through wet lab and now can be computationally estimated [55]. Similarly, the 16S rDNA analysis of VITHBRA024 showed 96% bootstrap alignment with *Bacillus paralicheniformis* in our previous study [32] and consistent results were seen in the phylogenetic tree obtained from aligning the whole genomes of type strains listed in TYGS database. The strain VITHBRA024 forms a clade with the *B. paralicheniformis* type strain KJ-16 with 71% bootstrap support (Fig 2B). The dDDH value of 93.2% with a confidence interval of 90.4–95.2% also confirms the identity of the strain as *B. paralicheniformis* species. A combined phylogenomic tree of both the candidate strains together, along with their type strains and *D. radiodurans* show the phylogenetic distance between the candidate strains (S1 Fig).

## Annotation of assembled genomes

A total of 5205 genes were detected by Prokka annotator for VITHBRA001 of which 5002 were CDSs along with 13 rRNA, 90 tRNA, 1 tmRNA and 99 misc_RNA. Through PGAP, the genome was detected to have 5092 genes and 4916 CDSs with 18 rRNA (5S-7, 16S-8, 23S-3), 91 tRNA and 6 ncRNAs while RAST SEED server provides 5188 coding sequences and 105 RNAs, of which 1695 proteins were recognised to be able to segregate them to different subsystems. In case of VITHBRA024, Prokka annotator detected 4410 genes of which 4254 are CDSs along with 4 rRNA, 56 tRNA, 1 tmRNA and 95 misc_RNA. PGAP detected 4340 genes of which 4196 were CDSs, along with 7 rRNA (5S-1, 16S-4, 23S-2), 53 tRNA and 5 ncRNA while RAST seed server provides with 4579 coding sequences and 60 RNAs and from coding sequences 1798 proteins could be segregated into different subsystems. All the annotators provide high quality output but each annotator uses different reference database and computational speed of annotation [56]. Resultantly, some of the distant genes that are marked "hypothetical" by one annotator could be identified with more precision by another annotator.

Further, to analyse the functional classification of the CDSs present in the candidate genomes under study, we used Cluster of orthologous groups (COG) database and specific metabolic pathways were identified using KEGG database. The results show that COG has classified both the genomes under 21 functional categories out of its 26 categories and could identify 75.81% of the CDSs for VITHBRA001 and 77.9% of the CDSs for VITHBRA024. Also, KEGG could provide metabolic pathway information on 52.7% of the CDSs of VITHBRA001

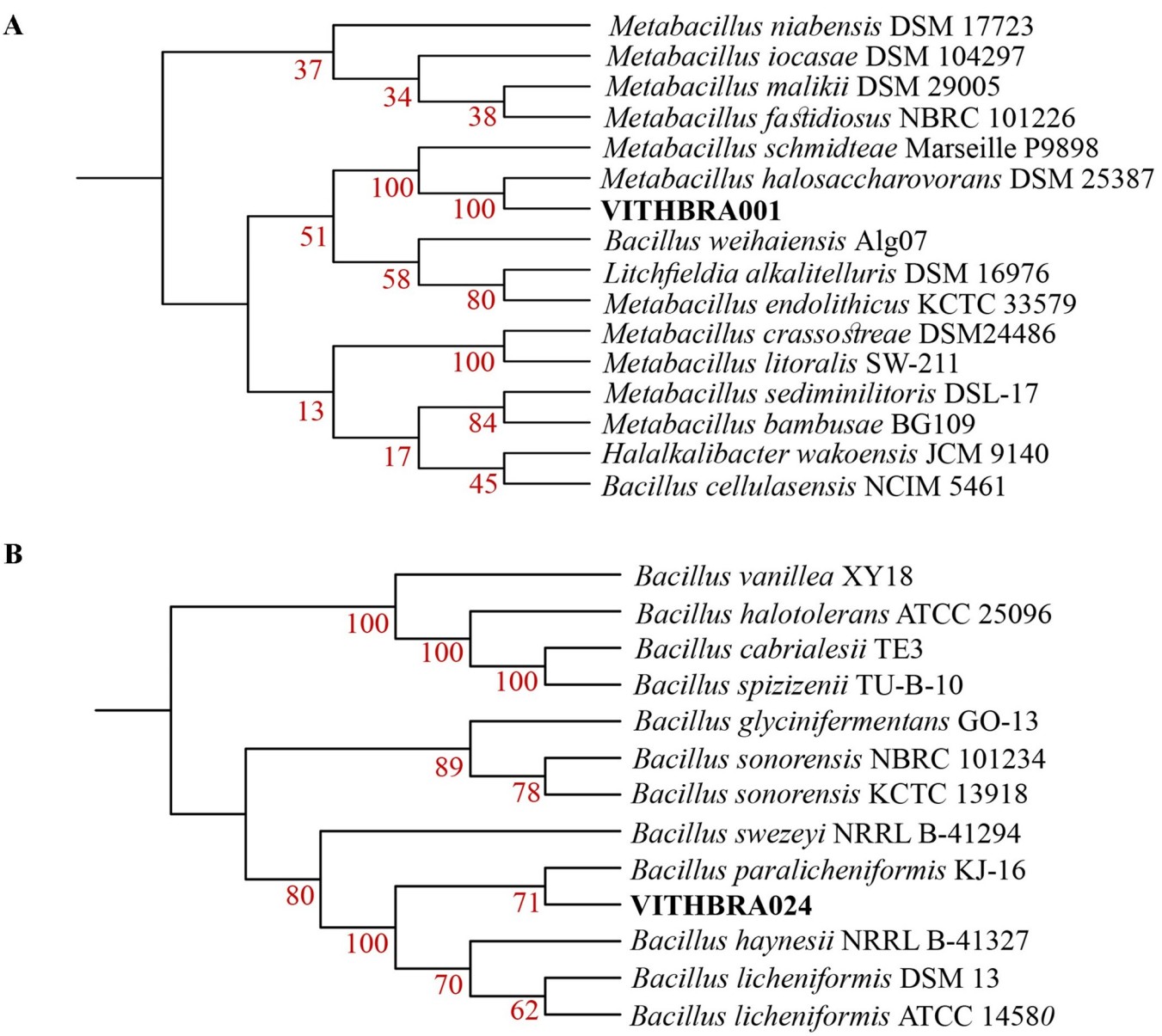

**Fig 2. Phylogenomics tree of VITHBRA001 and VITHBRA024.** In figure (A), the tree is constructed based on TYGS results which show the relatedness of strain VITHBRA001 with type strain organisms. With a bootstrap confidence of 100% (after calculating 100 replicates), it can be confirmed that VITHBRA001 belongs to the species *Metabacillus halosaccharovorans*. Similar process has been followed in figure (B), wherein the TYGS result shows the relatedness of strain VITHBRA024 with type strain organisms with a bootstrap confidence of 71% (after calculating 100 replicates), it can be confirmed that VITHBRA024 belongs to the species *Bacillus paralicheniformis*.

and 57.8% of CDSs of VITHBRA024. Again, the candidate strains show that almost 3.92% of the proteome has "unidentified function" as categorized under COG with an additional 24.19% of proteome not characterized by COG. Thus, total 28.11% of the VITHBRA001 proteome is unidentified. Further, VITHBRA024 has 4.02% of the proteome under "unidentified function" category of COG with an additional 22.10% of proteome not characterized by COG, making it a total of 26.12% of the proteome unidentified. These classification and annotation information show that much of the functional aspects of candidate genomes are unknown which could hold more knowledge of their radiation resistance property. A comparative

graphical representation has been done using both the organisms for COG and KEGG data
(Fig 3). It is observed that in COG classification, VITHBRA001 has greater number of
sequences under 17 categories as compared to four categories in VITHBRA024. The organisms
having higher number of functional genes under categories carbohydrate transport and metab-
olism (G), replication, recombination and repair (L), nucleotide transport and metabolism (F),
inorganic ion transport and metabolism (P), cell cycle control, cell division, chromosome par-
titioning (D) and cell wall/membrane/envelope biogenesis (M) have been reported to show
better resistance to radiation [57, 58]; this supports our experimental finding of the higher
resistance shown by VITHBRA001. Similar situation can be observed in case of KEGG data
wherein the parameters which encompass the genes involved in the above-mentioned COG
categories are higher in VITHBRA001 than VITHBRA024 which include nucleotide metabo-
lism, glycan biosynthesis and metabolism, genetic information processing, and environmental
information processing. The "signal transduction" category is also high in VITHBRA001 both
for COG and KEGG, which may represent better reception of environmental changes and
stress factors. Many "two component factors" such as DrRRA/DR2419, RadS/RadR, DrtS/
DrtR and DR1556/DRA0205 are studied to be important for *D. radiodurans* in their recovery
from radiation exposure [59]. Additionally, we can also observe through KEGG that our candi-
date strains have a significant number of genes in "unclassified pathway" categories for
"genetic information processing and signalling", and "cellular processes" as well as for "poorly
characterized" category suggesting possible presence of pathways that could help in radiation
resistance and even in these categories, VITHBRA001 has more genes as compared to VITH-
BRA024. This situation may assist VITHBRA001 to acquire better capability for resistance
than VITHBRA024.

## Possible genes responsible for radiation resistance

The present study has attempted, with the help of RAST and KEGG database, to manually
curate from the genome of the two candidate bacterial strains (VITHBRA001 and VITH-
BRA024), with a view to study the genes that are endogenous to them and the pathways that
could be responsible for radiation resistance (Fig 4). The annotation information of the
curated genes is in supplement information (S1 File).

**Homologous recombination (HR).**   KEGG pathway analysis (present study) suggests that
both the candidate strains (VITHBRA001 and VITHBRA024) could be dependent on RecFOR
pathway to efficiently repair the DSBs, a situation comparable with several radiation resistant
bacteria such as *D. radiodurans*, *R. radiotolerans* and *Cyanobacteria sp.* [12, 14, 60]. On the
other hand, the evident absence of *recB*, *recC* and *recD* genes in the candidate strains negates
the possibility for its dependence on an alternative RecBCD pathway which is reported to
repair DSBs in *E. coli*. Further, that RecBCD could enhance DSB repair in bacterial varieties or
not (in this case radiation resistant bacteria), is a question that deserves attention at this junc-
ture. Recombinant *D. radiodurans*, expressing *E. coli recBC*, showed increased sensitivity to
gamma radiation (to the tune of 2 log cycle reduction in resistance). This suggest that *recBC*
may not always have a positive role in DNA repair towards gamma resistance, thus advancing
our understanding on its functional limitation [61]. Table 2 lists all the genes that are endoge-
nous to both the candidate strains, and reported to be involved in RecFOR pathway. Previous
reports tempt us to suggest that, among the battery of genes, *recN* might be an early responder
when the cell in question encounters DSBs and form a repair center [62] wherein the basal
trimming and long end trimming proteins would subsequently function [63]. RecN is also
reported to be recruited after RecA and use its cohesion property in binding the sister chroma-
tids together during homologous recombination [64]. Further, either RecQ or RecS (homolog

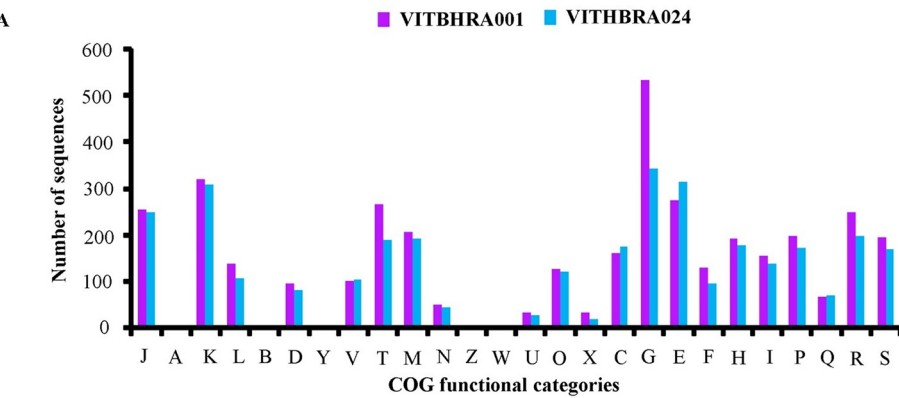

J: Translation, ribosomal structure and biogenesis

A : RNA processing and modification

K: Transcription

L : Replication, recombination and repair

B: Chromatin structure and dynamics

D: Cell cycle control, cell division, chromosome partitioning

Y: Nuclear structure

V: Defense mechanisms

T: Signal transduction mechanisms

M: Cell wall/membrane/envelope biogenesis

N: Cell motility

Z: Cytoskeleton

W: Extracellular structures

U: Intracellular trafficking, secretion, and vesicular transport

O: Posttranslational modification, protein turnover, chaperones

X : Mobilome: prophages, transposons

C : Energy production and conversion

G : Carbohydrate transport and metabolism

E: Amino acid transport and metabolism

F: Nucleotide transport and metabolism

H: Coenzyme transport and metabolism

I: Lipid transport and metabolism

P: Inorganic ion transport and metabolism

Q: Secondary metabolites biosynthesis, transport and catabolism

R: General function prediction only

S: Function unknown

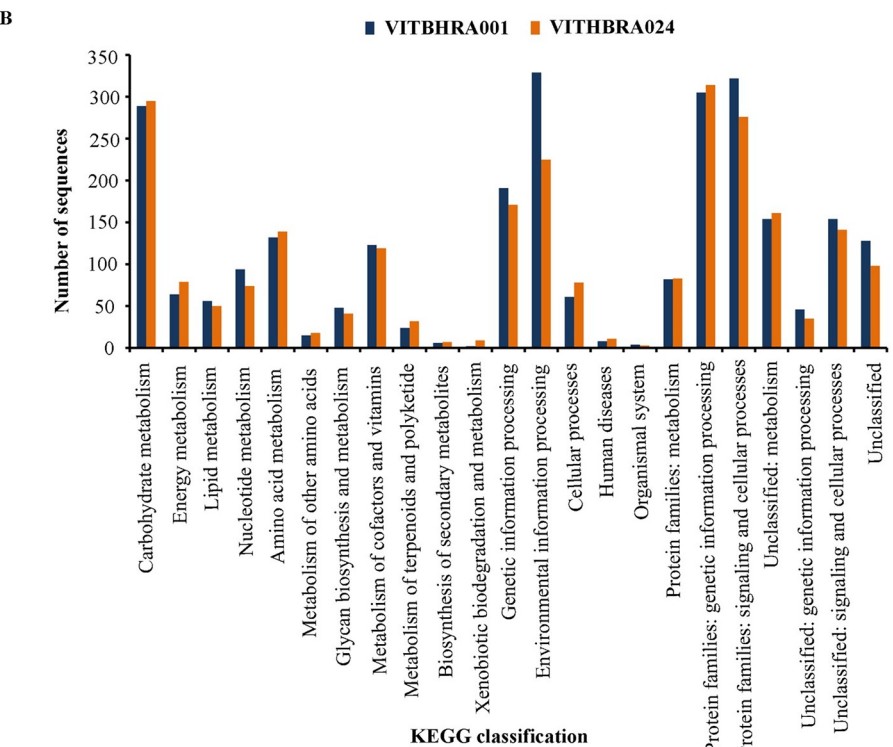

**Fig 3. Comparison of COG and KEGG data between VITHBRA001 and VITHBRA024.** Figure (A) shows the comparison of COG data and figure (B) shows comparison between KEGG data.

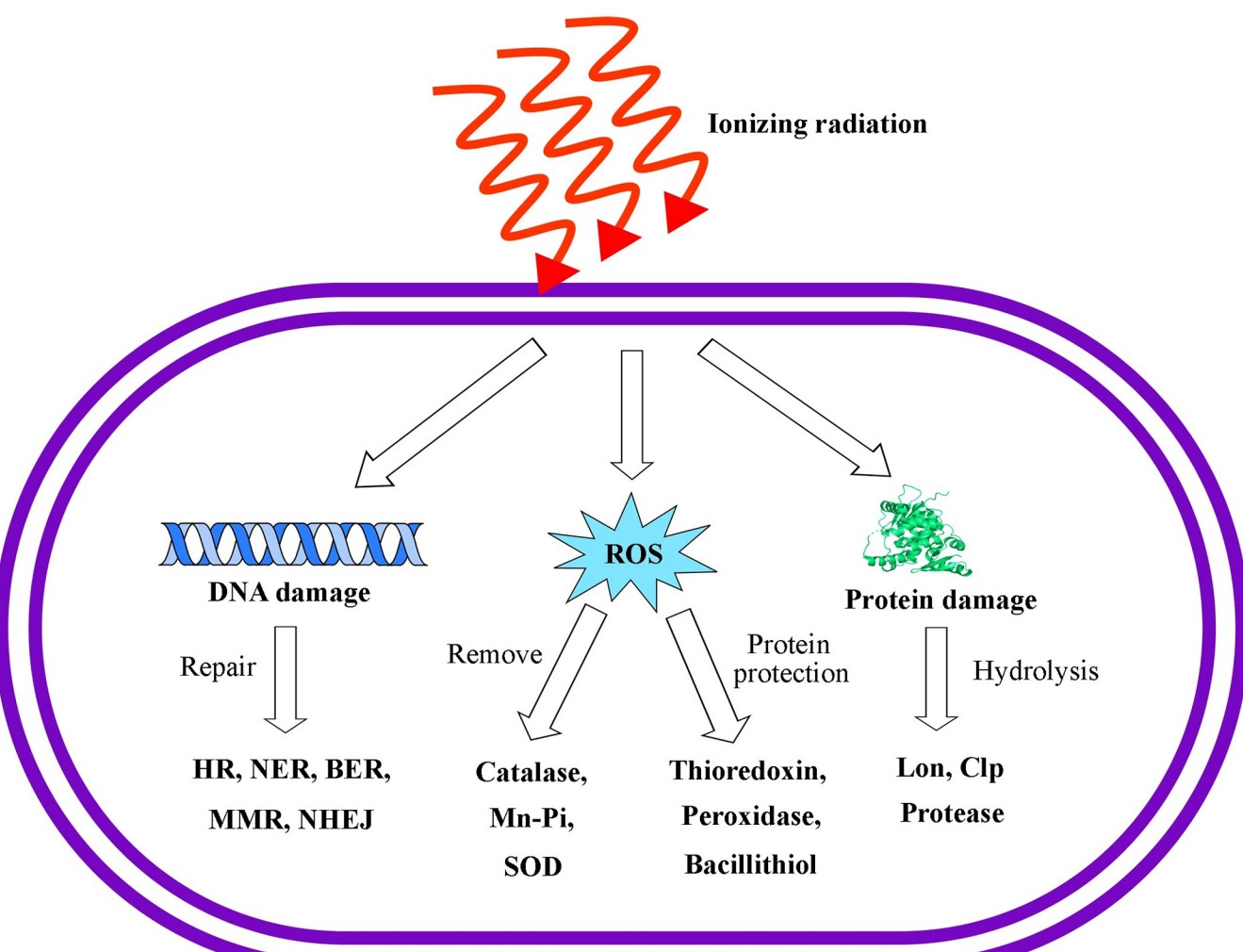

**Fig 4. Schematic representation of events in bacterial cell when exposed to ionizing radiation (IR).** The figure describes various damages that a cell undergoes due to IR exposure and the possible pathways to repair the damage.

of RecQ) helicases could be responsible for the long range end trimming in concert with RecJ exonuclease or through a complementary action of AddA helicase and AddB exonuclease [63, 65]. The staggered end ssDNA produced by exonucleases gets coated by SsbA proteins and help RecO, RecR, RecF, and RarA [66], to recruit the RecA proteins to result in the formation of the RecA-nucleofilament. Significantly, the presence of RecFOR, RecA and SsbA is crucial for the recovery of radiation resistant bacteria after irradiation, and absence of these sets of proteins makes the cell highly sensitive to irradiation [12, 67, 68]. RecA is also suggested to begin the search for homologous region with the help of RadA, and a D-loop intermediate is formed; RadA helps in unidirectional movement of the D-loop during homology search function [69]. Replication is restarted by the PriA-DnaB-DnaD proteins (primosome complex) which recruit the DnaC helicase on the single stranded DNA which in turn loads the DnaG primase and interacts with DnaX (tau protein subunit) to recruit the DNA polymerase C and DNA polymerase E [70]. The ligation after replication is completed with the help of DNA

**Table 2. Comparative list of homologous recombination repair genes present in the candidate strains, *D. radiodurans* and *E. coli*.**

| Common names of genes | Description | VITHBRA001 | VITHBRA024 | *D. radiodurans* | *E. coli* |
|---|---|---|---|---|---|
| *recN* | DNA repair ATPase RecN | 1 | 1 | 1 | 1 |
| *recJ* | DNA-specific exonuclease | 1 | 1 | 1 | 1 |
| *recQ* | DNA helicase RecQ | 1 | 1 | 1 | 1 |
| *recS* | RecQ-like DNA Helicase | - | 1 | - | - |
| *addA* | ATP dependent helicase/nuclease subunit A | 1 | 1 | - | - |
| *addB* | ATP dependent helicase/nuclease subunit B | 1 | 1 | - | - |
| *ssbA* | Single-stranded DNA-binding protein | 1 | 1 | 1 | 1 |
| *recB* | Exodeoxyribonuclease V beta subunit | - | - | - | 1 |
| *recC* | Exodeoxyribonuclease V gamma subunit | - | - | - | 1 |
| *recD* | Exodeoxyribonuclease V alpha subunit | - | - | - | 1 |
| *recO* | DNA repair protein RecO; Recombination protein O | 1 | 1 | 1 | 1 |
| *recR* | Recombinational DNA repair protein RecR | 1 | 1 | 1 | 1 |
| *recF* | Recombinational DNA repair ATPase RecF | 1 | 1 | 1 | 1 |
| *rarA* | Replication-associated recombination protein RarA | 1 | 1 | 1 | 1 |
| *recA* | Recombinase A | 1 | 1 | 1 | 1 |
| *radA* | DNA repair protein RadA | 1 | 1 | 1 | 1 |
| *recX* | Regulatory protein RecX | 1 | 1 | 1 | 1 |
| *uvrD* | DNA helicase | - | - | 1 | 1 |
| *pcrA* | UvrD-like helicase | 1 | 1 | - | - |
| *recG* | RecG-like helicase | 1 | 1 | 1 | 1 |
| *recD2* | ATP-dependent RecD-like DNA helicase | 1 | 1 | 1 | - |
| *ruvA* | Holliday junction resolvasome RuvABC, DNA-binding subunit | 1 | 1 | 1 | 1 |
| *ruvB* | Holliday junction resolvasome RuvABC, DNA helicase subunit | 1 | 1 | 1 | 1 |
| *recU* | Holliday junction resolvase | 1 | 1 | - | - |
| *ruvC* | Holliday junction resolvasome RuvABC, endonuclease subunit | 1 | - | 1 | 1 |
| *rusA* | Holliday junction resolvase | - | - | - | 1 |
| *mutS2* | dsDNA-specific endonuclease/ATPase | 2 | 1 | 1 | 1 |

polymerase I (DNA Pol I) and ligase A (LigA). The RecA nucleofilament extension is regulated by RecA modulators (anti-recombinase) which include RecX and PcrA, while the branch migration is facilitated by RecG translocase. Significantly, RecD2 helicase is also considered to play an important role along with RecG and/or RuvAB, as evidenced from induced double mutation experiments conducted by Torres *et al.* (2017) [71]. The branch migration forms the Holliday junction which is resolved into crossover (CO) and non-crossover (NCO) intermediates by a group of helicases such as RuvA, RuvB and endonuclease RecU. VITHBRA001 has two homologues of MutS2 (with 35.45% identity between them), one with a C-terminal SMR (small MutS related) region which has endonuclease activity (VITHBRA001_01182), an important function of this protein, while the other does not have the C-terminal SMR region (VITHBRA001_02893). The NCBI CD analysis suggests that VITHBRA001_02893 is indeed a MutS2 protein but the absence of SMR region makes it difficult to define its function (S2–S4 Figs). MutS2 is also observed to have diverse function in bacteria [72–74]. With a view to understand its possible function in the candidate strains, we compared the percentage identity of MutS2 from VITHBRA001_01182 and VITHBRA024_03392, with that of *D. radiodurans* (Dra) and *Bacillus subtilis* (Bsu) (S5 Fig), and could observe a high percentage of identity with MutS2$_{Bsu}$, which is reported to promote homologous recombination and could be recruited at Holliday junction or D-loop to act as endonuclease, similar to the function of RecU [75].

VITHBRA001 also has a *ruvC*-like gene (VITHBRA001_00722) confirmed by NCBI CD search analysis and Uniprot BLAST search (S6 Fig). The function of RuvC is to resolve Holliday junction and mainly produce NCO intermediates, similar to the function of RecU [76, 77]. Thus, VITHBRA001 seems to have three such genes (*recU*, *ruvC* and *mutS2*) which can help in resolution of Holliday junctions.

**Base excision repair (BER).** Multiple genes, particularly the glycosylases acting on a specific or broad range of substrates, are involved in BER pathway, enabling removal of inappropriately modified nucleotide bases, paving the way to open apurinic/apyrimidinic (AP) site [78]. Subsequent AP lyase and endonuclease activity would create 3′-OH and 5′-phosphate sites, leading to recruitment of DNA Pol I and LigA, to accomplish the repair. KEGG analysis of the WGS data reveals that VITHBRA001 has a total of 11 BER genes (Table 3). Almost similar situations seem to exist in the radiation resistant bacteria *H. taeanensis* (11 glycosylase genes), *D. radiodurans* (12 glycosylase genes) [79], *K. radiotolerans* (13 glycosylase genes) and *R. radiotolerans* (12 glycosylase genes) (KEGG database). VITHBRA024 have seven BER genes. *E. coli* also seem to have 11 genes but the most resistant organisms seem to have multiple copies of various glycosylase genes. Glycosylases act on the frequently formed nucleobase lesions caused by gamma radiation based oxidation including 8-oxo-7,8-dihydroguanine (8-oxoG), 2,6-diamino-4hydroxy-5-formamidopyrimidine (FapyGua), 5-hydroxycytosine (5-OHCyt), 5-hydroxyuracil (5-OHUra), 5,6-dihydroxy-5,6-dihydrouracil, 8-oxo-7,8-dihydroadenine (8-oxoA), 4,6-diamino-5-formamidopyrimidine (FapyAde) and thymine glycol (Tg) [80]. The base corrections are primarily being performed by formamido-pyrimidine-DNA glycosylase (*mutM/fpg*), A/G specific DNA glycosylase (*mutY*) and endonuclease III (*nth*). Significantly, the present WGS study reveals that all the afore-mentioned genes are present in both the candidate strains (Table 3). VITHBRA001 is shown to house two homologs of *mutM* (VITHBRA001_00999 and VITHBRA001_02212), known to encode a bifunctional glycosylase that removes bases along with phosphate backbone, specifically to damaged bases such as 8-oxoG, formamidopyrimidines, 5-OHCyt and 5-OHUra [81]. Further, one copy each of *mutY* (a monofunctional glycosylase responsible for removal of adenine (A) from 8-oxoG:A mismatch) and *nth* (responsible for removal of thymine glycol and prevent the cell from C to T or G to A transition mutation) is also present in both the candidate strains [82]. Importantly, a NUDIX protein MutT (nucleoside triphosphatase) which specifically converts the oxidized 8-oxo-dGTP to 8-oxo-dGMP is present in the candidate strains, so that this mutated

**Table 3. Comparative list of base excision repair genes in candidate strains, *D. radiodurans* and *E. coli*.**

| Common names of genes | Description | VITHBRA001 | VITHBRA024 | *D. radiodurans* | *E. coli* |
|---|---|---|---|---|---|
| *alkA* | 3-methyladenine DNA glycosylase II | 2 | 1 | 1 | 1 |
| *mpg* | 3-methyladenine DNA glycosylase | - | 1 | 1 | - |
| *tag* | 3-methyladenine DNA glycosylase | - | - | - | 1 |
| *ung* | Uracil DNA glycosylase | 1 | 1 | 1 | 1 |
| *udg4* | Uracil-DNA glycosylase | - | - | 1 | - |
| *mug* | G:T/U-mismatch repair DNA glycosylase | 1 | - | 1 | 1 |
| *mutY* | Adenine-specific DNA glycosylase, acts on AG and A-oxoG pairs | 1 | 1 | 1 | 1 |
| *mutM (fpg)* | Formamidopyrimidine-DNA glycosylase | 2 | 1 | 1 | 1 |
| *nth* | Endonuclease III | 1 | 1 | 3 | 1 |
| *nei* | endonuclease VIII | - | - | - | 1 |
| *xthA/exoA* | Exonuclease III | 1 | - | 1 | 1 |
| *nfo* | Endonuclease IV | 2 | 1 | - | 1 |
| *nfi* | Endonuclease V | - | - | 1 | 1 |

nucleotide does not get incorporated into the DNA. Endonuclease genes such as *nfo* and *xthA/exoA* act as endonucleases to finally create 3′-OH site for repair at the apurinic/apyrimidinic site [83]. VITHBRA001 has one *xthA* gene along with two *nfo* homologs (VITH-BRA001_00883 and VITHBRA001_02031) having percentage identity of 26.47% with each other, whereas VITHBRA024 has only one copy of *nfo*. Mutation of bases to hypoxanthin, uracil, 7-methylguanine and 3-methyadenine is counteracted by genes 3-methyadenine-DNA glycosylase (*mpg*, *alkA*) and uracil DNA glycosylase (*ung*, *mug*, *udg*). VITHBRA024 has two different 3-methyladenine DNA glycosylase (one belonging to each group of Mpg/Aag and AlkA) and VITHBRA001 has two homologs of AlkA (VITHBRA001_02631 and VITH-BRA001_02138) (S7 Fig). Aag (COG2094) is a eukaryotic type 3-methyadenine-DNA glycosylase whereas AlkA is prokaryotic type. Both seem to remove almost similar DNA base lesions such as hypoxanthin (a deaminated form of adenine), 7-methylguanine and 3-methyadenine as well as N-3 and N-7 adducts of purine and $O^2$ adducts of pyrimidine [84, 85]. The difference in specificity of these proteins (Aag and AlkA) towards base lesions may depend on the base mutations on ssDNA and dsDNA [86]. In order to specifically remove uracil (produced due to deamination of cytosine), a copy of *ung* (COG0692) gene is present in both the species of the present study. A copy of the *mug* (COG3663) gene is also present in VITHBRA001 which helps specifically remove uracil from uracil:guanine mismatches, preventing transition mutation [87]. The elevated expression of *mutY* and *ung* towards the mid stage of recovery from IR [12] and that of *alkA* during early stage of irradiation [88] encourage us to propose that these genes (Table 3) could be involved in radiation resistance in both the candidate species.

**Nucleotide excision repair (NER).** Expression of UvrABCD system (genes *uvrA*, *uvrB*, *uvrC*, *uvrD*) is known to facilitate removal of any DNA lesions that cause distortion in the DNA structure and integrity. Significantly, *uvrABC* genes are observed to be expressed at elevated levels in radiation resistant bacteria such as *D. radiodurans* and *K. radiotolerans* during recovery from ionizing radiation exposure [12, 15] and their presence is reported in various radiation resistant bacteria such as *Hymenobacter sp.* [19, 79, 89] along with other *Deinococcus sp.* [29, 90], strongly suggesting its importance in excision repair during IR irradiation. WGS analysis reveals the presence of UvrABC system (UvrA, UvrB, UvrC) in both the candidate strains (VITHBRA001 and VITHBRA024) (Table 4). UvrA and UvrB are known to recognise the damage, and recruit UvrC which further cleaves the two ends of the damaged region. UvrD—a helicase—removes this cleaved fragment and allows DNA Pol I and LigA to repair the damage. VITHBRA001 has two homologs of UvrA (VITHBRA001_00376 and VITH-BRA001_02130) (UvrA1 and UvrA2 respectively), one copy of UvrB and two copies of UvrC (VITHBRA001_01190 and VITHBRA001_01609), and additionally, two homologues of UvsE (VITHBRA001_00228 and VITHBRA001_01817), but not UvrD. A comparable situation of

**Table 4. Comparative list of Nucleotide excision repair genes in candidate strains, *D. radiodurans* and *E. coli*.**

| Common names of genes | Description | VITHBRA001 | VITHBRA024 | *D. radiodurans* | *E. coil* |
|---|---|---|---|---|---|
| *uvrA1* | Excinuclease UvrABC, ATPase subunit | 1 | 1 | 1 | 1 |
| *uvrA2* | Excinuclease UvrABC, ATPase subunit | 1 | - | 1 | - |
| *uvrB* | Excinuclease UvrABC, helicase subunit | 1 | 1 | 1 | 1 |
| *uvrC* | Excinuclease UvrABC, nuclease subunit | 1 | 1 | 1 | 1 |
| *uvrD* | DNA helicase | - | - | 1 | 1 |
| *pcrA* | UvrD-like helicase | 1 | 1 | - | - |
| *uvsE* | UV DNA damage repair endonuclease | 2 | - | 1 | - |
| *Cho* | excinuclease | - | - | - | 1 |

two homologues of UvrA is reported to exist in *D. radiodurans*; UvrA1 is demonstrated to be essential for radiation resistance in NER pathway, as it has the UvrB interaction domain that is absent in UvrA2 [91]. Among the two UvrA homologues of VITHBRA001, VITH-BRA001_00376, could be comparable with UvrA1 of *D. radiodurans*, with 57.38% identity, and also due to the presence of UvrB interaction domain (observed from NCBI CD search analysis) (Fig 5). With the absence of UvrB interaction domain, VITHBRA001_02130 could be comparable with UvrA2 (Fig 6). It is reported that UvrA2 is present in many bacteria obtained from harsh environment and it may function in NER in the absence of UvrA1 but its mode of action is elusive [91, 92]. VITHBRA024 has only a single copy of each gene of UvrABC system, but not UvrD. Among the two copies of UvrC in VITHBRA001, VITHBRA001_01190 seems to be the authentic UvrB interacting gene, thus this can be stated as UvrC1 (Fig 7); however, VITHBRA001_01609 (considering as UvrC2), has the N-terminal GIY-YIG domain (observed by NCBI CD search analysis of the present study), responsible for the 3′ incision reaction in NER pathway (Fig 8). The deficiency due to the absence of UvrD in both the strains could apparently be offset with the presence of UvrD-like helicases. One such helicase PcrA is reported to take the role of UvrD [93] which might be the situation in our strains. Additionally, the occurrence of two homologs of UvsE in VITHBRA001, which have 36.59% percentage similarity with each other, draws our attention, as it appears to be comparable with the existence of a UvsE-dependent (UVER) excision pathway of *D. radiodurans*. It is also reported that UvsE could partially replace the function of UvrA1 when it is knocked out of the cell [91, 94]. *E. coli* is observed to be devoid of *uvrA2* and *uvsE* genes.

**Mismatch repair (MMR).** MMR is known to help remove any wrong incorporation of nucleotide during replication. *MutS* and *MutL* genes are shown to play pivotal role in MMR wherein the MutS gets recruited at the site of mismatch which in turn recruits MutL. In contrast to the presence of MutH in *E. coli* which causes the incision at the site of mismatch, MutL does the function [95]. The presence of these genes in the candidate strains (Table 5), prompts us to suggest for their dependence on this pathway. Pertinently, MMR is reported to be one of the key pathways in extreme radiation resistance in *D. radiodurans*, wherein *MutS* gene is observed to be significantly expressed during the period of its recovery from irradiation [88].

**Non homologous end joining (NHEJ) repair.** The candidate strains would also depend on NHEJ pathway, a requisite for the repair of DSB caused by IR at quiescent or stationary phase, judged from the genomic presence of Ku and LigD, the primary proteins involved in repair [96] (Table 5). Additionally, CD search tool of the present study reveals the bifunctional nature of LigD, with polymerase and ligase domains (S8 Fig), but without the phosphoesterase domain, comparable to the situation in LigD$_{Bsu}$ [97]. Further, that the lack of phosphoesterase domain in LigD could be offset by pnpase (*pnp*) which could help in removal of any 3′-PO$_4$ at the DSBs to 3′-OH for ligase to act on, has been suggested in previous investigations [98]. *D. radiodurans* also seem to have NHEJ pathway (though not Ku and LigD), through a novel *pprA* gene which bind to the DSBs and help recruit ligase to fix the damage [99, 100]. *E. coli* does not have NHEJ pathway but alternate end joining pathway (A-EJ) which requires LigB and repairs DNA mainly with deletion mutations. The process is a last resort which occurs only if HR was unable to repair DSBs [101].

**Nucleotide diphosphate with X moiety (Nudix).** Nudix proteins are known to help remove the damaged nucleotides such as nucleotide diphosphates derivatives from the cellular system, so that they are not mis-incorporated into the DNA sequence to cause error. In case of VITHBRA001 and VITHBRA024, we could manually curate twelve and seven nudix proteins, respectively (Table 6). As mentioned in the above BER section, MutT nudix protein is required to specifically remove the oxidatively damaged product 8-oxo-dGTP and 8-oxo-GTP. Using NCBI CD search, we could identify a copy of its prototype homolog in VITHBRA001 (00018)

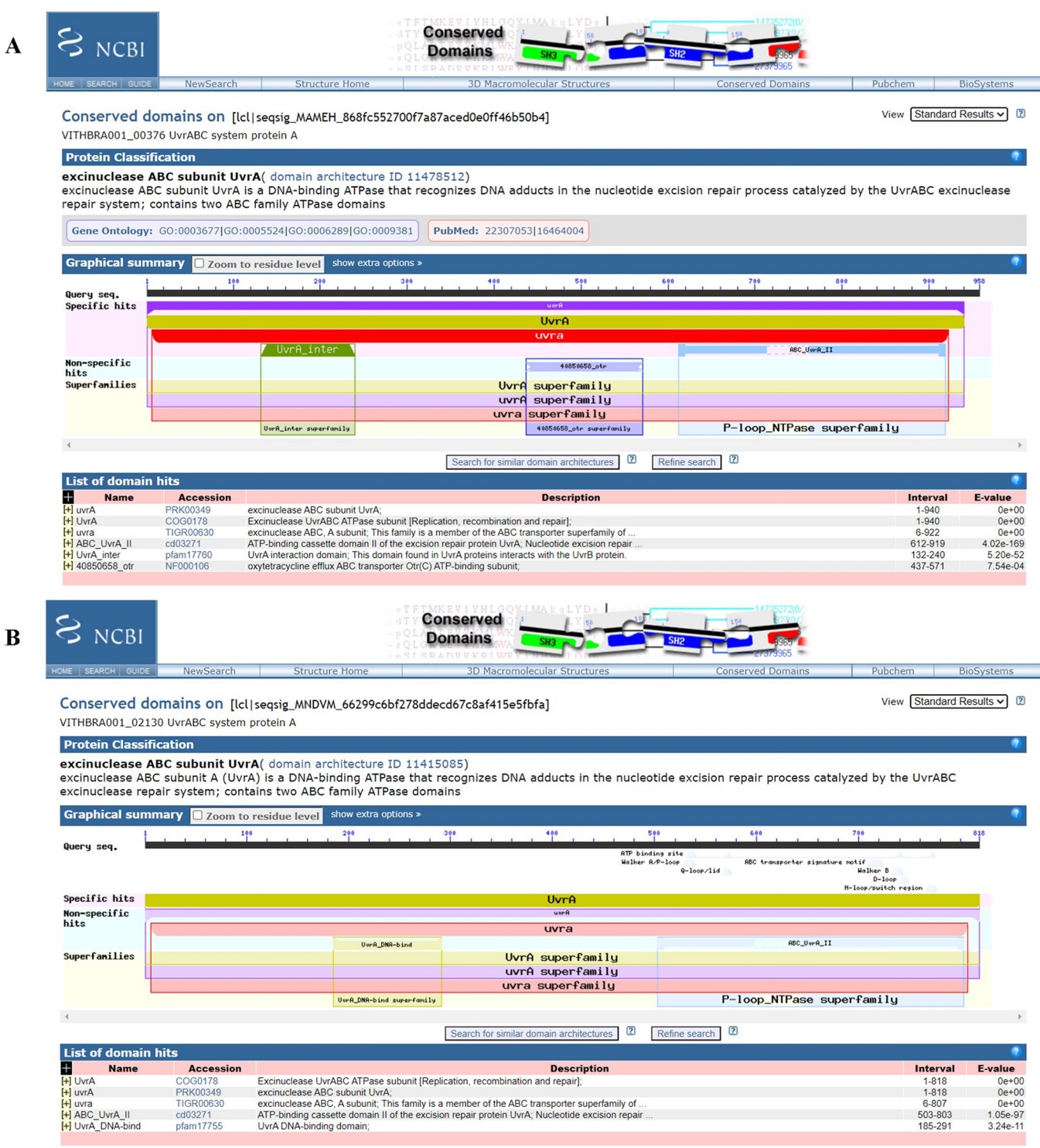

**Fig 5. NCBI CD search analysis of VITHBRA001_00376 and VITHBRA001_02130.** In figure (A) CD search analysis identifies the presence of 'UvrA_inter' region (shown in green) in VITHBRA001_00376, which is responsible for interacting with UvrB during the nucleotide excision repair (NER), while in figure (B), CD search identifies UvrA domain in VITHBRA001_02130 but without the UvrB interacting region. Thus, this is annotated as UvrA2 which might not have the conventional UvrA function in the NER pathway.

```
VITHBRA001_00376    MAMEHIVVKGARAHNLKNIDVTIPRDKLVVVTGLSGSGKSSLAFDTIYAEGQRRYVESLS    60
VITHBRA001_02130    -MNDVMIVKGAKENNLKDLSVSIPKNKLIVVTGPSGSGKSTLAMDTLFKECQRQYLESMG    59
                     : ::****: :***::.*:**::**:**** ******:**:**:: * **:*:**:.

VITHBRA001_00376    AYARQFLGQMDKPDVDAIEGLSPAISIDQKTTSRNPRSTVGTVTEIYDYLRLLYARVGRP   120
VITHBRA001_02130    ------LQGINKPKVDSISGLSPAISINQQNTNRNPRSTVGTVTDMYTSLRMVFEKLGKR   113
                          *  ::**.**:*.*********:*:.*.************::*  **::: ::*:

VITHBRA001_00376    TCPVHGIEISSQTIEQMVDRILEYPERTKLQVLAPIVSGRKGTHVKVFEDIKKQGYVRVR   180
VITHBRA001_02130    QCPACDHDVDPTFNT---------------------------------------------   128
                     **. . ::.

VITHBRA001_00376    VDGEMHELSEEIELEKNKKHSIEVVIDRIVVKEGVASRLADSLEAALGLGEGRVIIDVMG   240
VITHBRA001_02130    --------------------------------------------------------EDIE   132
                                                                             : :

VITHBRA001_00376    EEELLFSEHHACPQCGFSIGELEPRMFSFNSPFGACPECDGLGSKLKVDLDLVLPNKDLS   300
VITHBRA001_02130    KEEGSFKEYIFCPNCHHRMEKLTRTHFSYNTTEGACSTCKGLGETVDIHKESVF-HQHLS   191
                    :** *.*:  **:* .: :*    **:*: *** *.***..:.:. : *: ::.**

VITHBRA001_00376    LKQHAIAPWEPTSSQYYPQMLEAVCNHFGIDMD--IPVKDIPKHLLDKVLYGSDGEEIYF   358
VITHBRA001_02130    LEDGAVDLWKGRYADYQIPVVKAAMAYYGVPIEDGLPMQEYNPVQKAILYYGVESDEVKG   251
                    *:: *: *:  ::*   :::*.  ::*: :: :*::: : ** :.:*:

VITHBRA001_00376    RYENDFGQVRENYIQFEGVIRNVERRYKETTSDFIREQMEKYMGQQNCPSCKGYRLKKET   418
VITHBRA001_02130    YFPDKKAPKTVDKGKFEGVLTGMWRRFTEKSGS--SSEAEEYFYSQVCPDCHGERLNEIS   309
                     : :.     :   :****: .: **:.*.:..   .: *:*: .* **.*:* **:: :

VITHBRA001_00376    LAVLIQGHHVGEITKLSVQESLEFFQNLSL--TEKEMQIANLILKEICERLGFLNNVGLD   476
VITHBRA001_02130    RSVTVENMTIPKLVSHSLEDMLKWAEQLEKGLDKVSYMLVETFLQDMKTKLTRIIKIGLG   369
                     :* ::. : ::. *::: *:: ::*. : .  ::*::: :* : ::**.

VITHBRA001_00376    YLTLNRAAGTLSGGEAQRIRLATQIGSRLTGVLYILDEPSIGLHQRDNDRLIQTLQNMRD   536
VITHBRA001_02130    YLTLDRQIITLSGGETQRLRLSALLDSALTGVLYIMDEPTVGLHPKDTLGLVSVLKGLRD   429
                    ****:*    ******:**:**:: :.* *******:***::*** :*.  *::.*:.:**

VITHBRA001_00376    IGNTLIVVEHDEDTMMAADYLIDIGPGAGIHGGEVISAGTPEEVMNDDNSLTGQYLSGKK   596
VITHBRA001_02130    LGNTVLLIEHDVDVMKEADYIIDIGPGAGKLGGTVVGQGTLQELKEQESSVTGRYLREEE   489
                    :***:::*** *.* ***:********  ** *:. ** :*: :::.*:**:** ::

VITHBRA001_00376    FIPLPYERKKPDGRYIEIKGAKENNLRNVSVKFPLGTFIAVTGVSGSGKSTLVNEILHKT   656
VITHBRA001_02130    EMDRTYR--KGTGQAISVHHATIHNLKDITVTFPIGCLTAVTGVSGSGKSSLVFDVLAKG   547
                     :  *.  *  *: *.:: *. :**:::*.**:*. ************:** ::* *

VITHBRA001_00376    LAQKLHKAKSKPGEHKGVKGIEHLEKVIDIDQSPIGRTPRSNPATYTGVFDDIRDVFATT   716
VITHBRA001_02130    NEK-------IHDGFDRVTGLDHFDQMIIVGQSPLSRMKRSNIATYIDVFTHIRTIFSKD   600
                     :          . .. *.*::*:::* :.***:.* *** *** .** .** :*:.

VITHBRA001_00376    NEAKVRGYKKGRFSFNVKGGRCEACRGDGIIKIEMHFLPDVYVPCEVCHGKRYNRETLEV   776
VITHBRA001_02130    KAAKEKGLTAKHFSFNTVGGRCENCQGLGYVTTNMLFFPDLEVVCPVCQGKRFQKDVLSI   660
                    : ** :* .   :****. ***** *:* *  *  :* *:** * * **:***::::.*.:

VITHBRA001_00376    TYKGKNISDVLEMTVEDAVSFFENIPKIKRKLQTIYDVGLGYITLGQPATTLSGGEAQRV   836
VITHBRA001_02130    KYNDHSVNDILESSIVDCLSIFENEKKVKEVLELLVEIGLGYLKMGQSLTTLSGGEGQRL   720
                    .*:.::.:.*:** :: *.:*:*** *:*. *: : ::****:.:** *******.**:

VITHBRA001_00376    KLASELHRRSSGRSLYIILDEPTTGLHVDDIARLLKVLQRLVDNGDTVLVIEHNLDVIKAT   896
VITHBRA001_02130    KLAKELLKQGNKNSLYLLDEPTTGLHPNDVTQLLKLLNRLVDAGHTVIMVEHNSQMIKGA   780
                    ***.** ::.. .***:*:********* :.* *:**:**:.*:**:::**** ::**.:

VITHBRA001_00376    DYLVDLGPEGGDKGGQIVGYGTPEDIMNNEQSYTGKYLKPVIERDRDRMRKLIKEKEEVA   956
VITHBRA001_02130    DWIIDLGPEGGGKGGQITAQGTPESIIKNSNSYTGQYL----------------------   818
                    *::.*******.*****.. ****.*::*.:****:**

VITHBRA001_00376    QS       958
VITHBRA001_02130    --       818
```

**Fig 6. Multiple sequence alignment (MSA) of two homologs of UvrA.** The highlighted section in this figure shows the absence of UvrB interacting region in VITHBRA001_02130 as compared to VITHBRA001_00376. The MSA was performed using Clustal omega version 1.2.2.

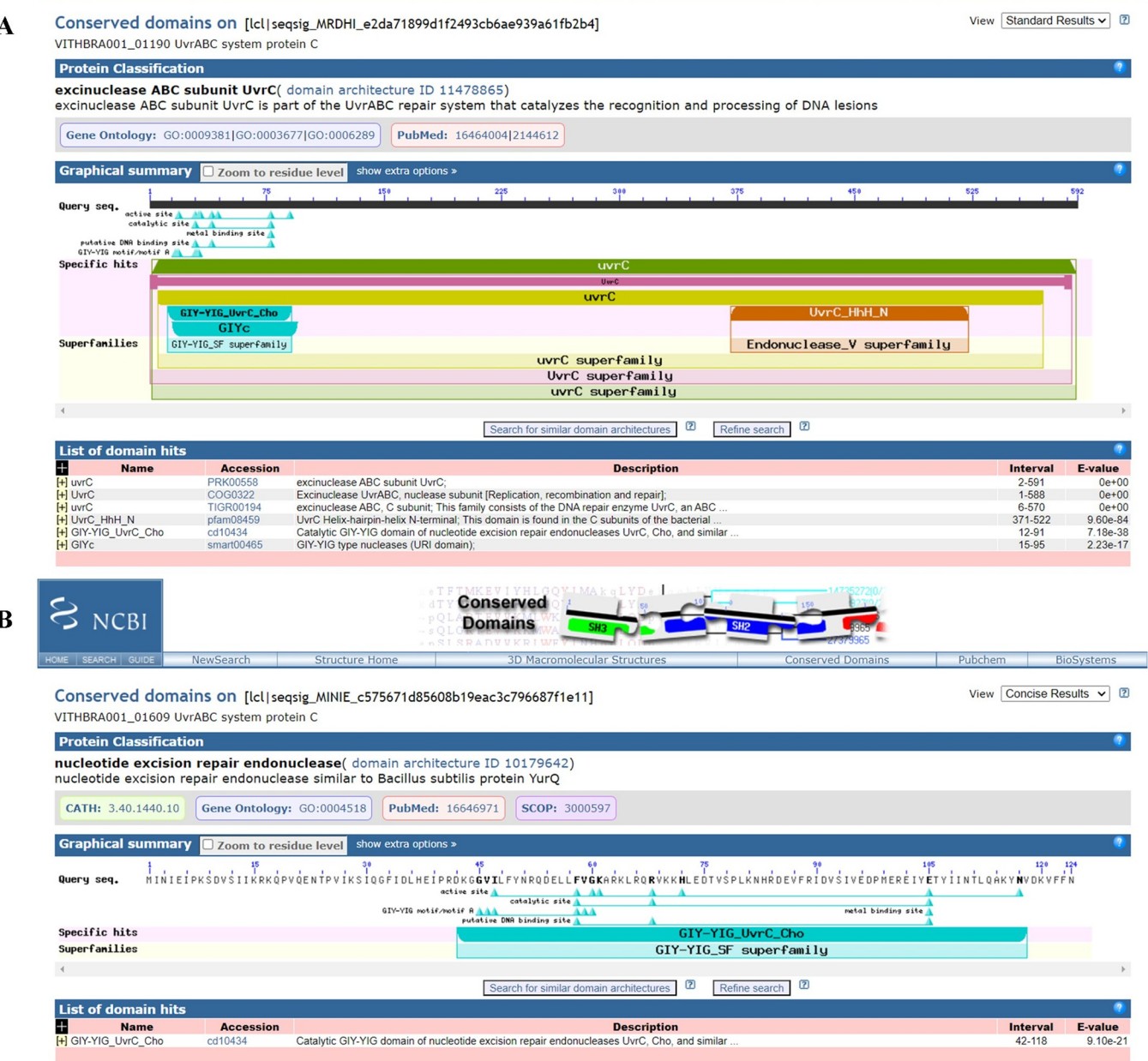

**Fig 7. NCBI CD search analysis of VITHBRA001_01190 and VITHBRA001_01609.** In figure (A) VITHBRA001_01190 is identified as full-length UvrC protein while in figure (B), VITHBRA001_01609 is identified as a short protein having only the N-terminal region of UvrC protein which is responsible for the 3′ incision reaction in the UvrABC system of NER pathway.

(S9 Fig) which shared 31.78% identity with *E. coli* MutT, but this gene is not present in VITH-BRA024. Additionally, a MutT mammalian homolog is present in both the strains (VITH-BRA001_04257 and VITHBRA024_01609) which does the same function of removing 8-oxo-dGTP/8-oxo-GTP [102] along with the presence of a putative MutT homolog in both the strains (VITHBRA001_05141 and VITHBRA024_04127). VITHBRA001 also possesses two more genes belonging to MutT family. It is reported that *D. radiodurans* has a total of 23 Nudix proteins and during the early stage of radiation recovery it strongly induces MutT

```
VITHBRA001_01609    MINIEIPKSDVSIIKRKQPVQENTPVIKSIQGFIDLHEIPRDKGGVILFYNRQDELLFVG    60
VITHBRA001_01190    --------------------------MRDHIKEKLALLPDQPGCYIMKDRQGTVIYVG    32
                                              : : :    :   *: *   :: :**. :::**

VITHBRA001_01609    KARKLRQRVKKHLEDTVSPLK-NHRDEVFRIDVSIVEDPMEREIYETYIINTLQAKYNVD   119
VITHBRA001_01190    KAKVLKNRVRSYFTGSHDGKTLRLVNEITDFEYIITSSNLEALILELNLIKKYDPKYNVM    92
                    **: *::**:.:: .: .  .  . :*:  :: *...:*  * *  :*:. : ****

VITHBRA001_01609    KV------FFN-------------------------------------------------   124
VITHBRA001_01190    LKDDKTYPFIKITNELHPRLLVTRQVKKDKGKYFGPYPNVQSARETIKLLDRLYPLRKCS   152
                            *::

VITHBRA001_01609    ------------------------------------------------------------   124
VITHBRA001_01190    TLPDRVCLYYHMGQCLAPCVNDVSEETNRQMVEEISKFLNGGYKGIKEELSIKMTKAAEG   212

VITHBRA001_01609    ------------------------------------------------------------   124
VITHBRA001_01190    LEFERAKEFRDQILHIEATMEKQKMTLNDFIDRDVFGYAYDKGWMCVQVFFIRQGKLIER   272

VITHBRA001_01609    ------------------------------------------------------------   124
VITHBRA001_01190    DVSMFPIYDNPEEEFLTFLGQFYSKSNHFLPREILLPDSIEFDLVEQLLDVTTLQPKRGK   332

VITHBRA001_01609    ------------------------------------------------------------   124
VITHBRA001_01190    KKDLILLAHKNAKIALKEKFLLIERDEERTIKAVENLGDKLGIHTPNRIEAFDNSNIQGT   392

VITHBRA001_01609    ------------------------------------------------------------   124
VITHBRA001_01190    DPVSAMVVFEDGKPAKKHYRKYKIKDVKGPDDYDSMREVVRRRYSRVLKEQLPLPDLIII   452

VITHBRA001_01609    ------------------------------------------------------------   124
VITHBRA001_01190    DGGKGHLSAAQDVLENELGLDIPVAGLVKDDKHRTSELIIGSPPEFIQLERNSQEFYLLQ   512

VITHBRA001_01609    ------------------------------------------------------------   124
VITHBRA001_01190    RIQDEVHRFAITFHRQLRGKNAFQSILDDIPGVGEQRKKSLLKHFGSVKKMKEASVDDLK   572

VITHBRA001_01609    -------------------- 124
VITHBRA001_01190    RAGMPDNIANNILEHLHKES 592
```

**Fig 8. Multiple sequence alignment of two homologs of UvrC protein.** The highlighted region in this figure represents the region having 3′ incision function in both the UvrC proteins. The MSA was performed using Clustal omega version 1.2.2.

(DR0261) along with four more nudix proteins [12]. A recent study in mammalian cell suggests that ADP-ribose hydrolase of NUDT5 motif helps in DNA repair in homologous recombination pathway as well as supply with ATP required for repair by hydrolysing ADP-ribose [103]. Using NCBI CD search we could find that each of the strains possesses an ADP-ribose hydrolase of NUDT5 motif (VITHBRA001_04428 and VITHBRA024_02687) which could also work for DNA damage repair in these strains.

## Protection from oxidative stress

Production of ROS causes indirect harm to the proteins present in the cell and disrupts their function. Cell has various mechanisms to remove ROS from the system and protect the proteins from ROS (Fig 4).

**Catalases.** The catalase family of enzymes remove hydrogen peroxide from the system, which if present in the system will react with the cellular molecules to cause damage and loss of function. VITHBRA001 and VITHBRA024 both have three monofunctional heme binding catalases (Table 7) each belonging to three separate clades [104]: Catalase-clade I/KatX

**Table 5. Comparative list of mismatch repair genes and NHEJ genes in candidate strains, *D. radiodurans* and *E. coli*.**

| Common names of genes | Description | VITHBRA001 | VITHBRA024 | *D. radiodurans* | *E. coil* |
|---|---|---|---|---|---|
| *mutL* | DNA mismatch repair ATPase | 1 | 1 | 1 | 1 |
| *mutS* | DNA mismatch repair ATPase | 1 | 1 | 1 | 1 |
| *mutH* | DNA mismatch repair protein MutH | - | - | - | 1 |
| *dam* | Site-specific DNA-adenine methylase | - | - | - | 1 |
| *vsr* | DNA mismatch endonuclease | - | - | - | 1 |
| *ku* | Non-homologous end joining protein Ku | 1 | 1 | *pprA* gene specific to *D. radiodurans* | - |
| *ligD* | Bifunctional non-homologous end joining protein LigD | 1 | 1 | | - |

(VITHBRA001_03351 and VITHBRA024_02002), Catalase-clade II/ KatE (VITH-BRA001_01560 and VITHBRA024_02070) and Catalase-clade III/KatA (VITHBRA001_01472 and VITHBRA024_02003). In *D. radiodurans*, clade-I/clade II catalase is reported to be requisite for resistance against acute and/or chronic IR [105]. In addition to the above group of catalases, the candidate strains also encode manganese catalases (MnKat) which bind Mn in place of heme group. Judged from Prokka analysis, VITHBRA024 has three types of MnKat genes and VITHBRA001 houses four types of MnKat, out of which one type is designated as spore coat protein with Mn-catalase activity (VITHBRA001_03891 and VITHBRA024_00264), confirmed by RAST and UNIPROT analyses (Table 7). Mn-catalases are mainly reported to be abundant in Deinococcus-Thermus and Firmicutes [106], though it is absent in *D. radiodurans*; a single copy is, however, present in most of the other *Deinococcus* species [29]. *E. coli* (Eco) is also deprived of MnKat. Significantly, Mn-catalases have been reported to be found in organisms inhabiting extreme environments. It is estimated that Mn-catalases could be functional against low levels of $H_2O_2$ and it could function in cells having iron limitation [107]. Thus Mn-catalases can have high potential to function against $H_2O_2$ produced due to IR exposure in limited quantity such as in HBRA and protect the cells from protein oxidation.

**Superoxide dismutase (SOD).** SODs are metalloenzymes which remove superoxide radicals and produce $H_2O_2$ and $O_2$. The strains VITHBRA001 and VITHBRA024 consist of three

**Table 6. Comparative list of Nudix genes in the candidate strains, *D. radiodurans* and *E. coli*.**

| Common names of genes | Description | VITHBRA001 | VITHBRA024 | *D. radiodurans* common genes | *E. coli* |
|---|---|---|---|---|---|
| *mutX* | 8-oxo-dGTP diphosphatase | 1 | 1 | 8 | - |
| *ytkD* | Putative 8-oxo-dGTP diphosphatase YtkD | 1 | 1 | | - |
| | 7,8-dihydro-8-oxoguanine-triphosphatase | 1 | - | | 1 |
| *mutT4* | Putative Mutator MutT4 | 1 | - | | - |
| | MutT superfamily | 1 | - | | - |
| *NudF* | ADP-ribose pyrophosphatase (NUDT5) | 1 | 1 | 3 | 1 |
| | Putative ADP-ribose pyrophosphatase | 3 | 4 | | |
| *nudC* | NADH pyrophosphatase | 2 | - | 1 | 1 |
| *idi* | Isopentenyl-diphosphate Delta-isomerase | 1 | - | - | - |
| *dut* | dUTP diphosphatase | - | - | - | 1 |
| *nudB* | Dihydroneopterin triphosphate diphosphatase | - | - | - | 1 |
| *nudL* | putative Nudix hydrolase NudL | - | - | 1 | 1 |
| *nudJ* | Phosphatase NudJ | - | - | 1 | 1 |
| *nudI* | Nucleoside triphosphatase NudI | - | - | | 1 |
| *yfcD* | putative Nudix hydrolase YfcD | - | - | 1 | 1 |
| *nudK* | GDP-mannose pyrophosphatase NudK | - | - | | 1 |

**Table 7. Comparative list of genes with catalase activity in candidate strains, *D. radiodurans* and *E. coli*.**

| Common names of genes | Description | VITHBRA001 | VITHBRA024 | *D. radiodurans* | *E. coli* |
|---|---|---|---|---|---|
| *katX* | monofunctional heme catalase- clade-I | 1 | 1 | 1 | - |
| *katE* | monofunctional heme catalase- clade-II | 1 | 1 | 1 | 1 |
| *katA* | monofunctional heme catalase- clade-III | 1 | 1 | - | - |
| *katG* | bifunctional heme catalase-peroxidase | - | - | - | 1 |
| MnKat | Mn- containing catalase | 4 | 3 | - | - |

SOD genes: *sodA* (Mn-binding), *sodF* (Fe-binding) and *sodC* (Cu/Zn-binding) (Table 8). With respect to identity of SOD genes between the two strains, the percentage of identity for SodA, SodF and SodC is 82.18%, 56.99% and 40.23% respectively. SodA of VITHBRA001 shows 69.15% identity with that of $SodA_{Dra}$ and 63.96% with $SodA_{Eco}$, while VITHBRA024 shows 64.68% identity with $SodA_{Dra}$ and 60.91% with $SodA_{Eco}$. SodA of *D. radiodurans* is reported to be produced constitutively in high levels, helping in its radiation resistance. SodC of both the candidate organisms shows around 30% identity with the two SodC of *D. radiodurans* as well as with *E. coli*. SodF is absent in *D. radiodurans* and other *Deinococcus* species, suggesting less requirement of iron binding proteins in *D. radiodurans* for its resistance to radiation [29].

**Peroxidases.** Evidenced from WGS, both the candidate strains possess the genes such as *bcp* (bacterioferritin comigratory protein), *tpx* (thiol peroxidase) and *ahpC* (alkyl hydroperoxide reductase) (Table 9) which are known to remove peroxides from the system, but in-turn themselves get oxidized to produce intramolecular disulphide bonds. The oxidized states (peroxides) are then converted to their reduced form with the help of thioredoxin system and AhpF protein, which works with AhpC. AhpC is present in two homologs in each strain, wherein one is typical 2-cys containing (VITHBRA001_03582 and VITHBRA024_00675) while the other is 3-cys containing (VITHBRA001_01696 and VITHBRA024_02173) (S10 Fig). Although both the AhpC homologs are invoked against oxidative stress, 2-cys AhpC is apparently more sensitive to $H_2O_2$ even in low concentration as compared to 3-cys AhpC which is very less sensitive even in high levels of $H_2O_2$. Further, 3-cys AhpC is effective against lipid hydroperoxide and was induced when exposed to organic peroxides [108]. Thus, presence of two AhpC would make the candidate strains efficient against ROS in various concentrations. On the other hand, *E. coli* is observed to have one 2-cys AhpC gene along with a copy of AhpF whereas most of the *Deinococcus* species do not possess AhpCF. *D. radiodurans* mainly have AhpE (a 1-cysteine homolog to AhpC) and AhpD (functionally similar to AhpE), wherein AhpD mutated strain of *D. radiodurans* is reported to show increased sensitivity to $H_2O_2$ stress [109].The Bcp protein (one copy each) present in VITHBRA001 and VITHBRA024 have 43.87% and 41.67% identity respectively to DR0846 (one of the three homologs of Bcp) of *D. radiodurans* and < 30% identity with the other two homologs of Bcp of *D. radiodurans* (S11 Fig). DR0846 is reported to express under IR stress [110] and the *bcp* gene in general is activated against oxidative stress in various bacteria [111–113]. The substrate specificity of Bcp largely remain unknown [110] but one report suggests that it has more specificity towards fatty acid hydroperoxides rather than $H_2O_2$ and organic peroxides [111].

**Table 8. Comparative list of genes with superoxide dismutase function in candidate strains, *D. radiodurans* and *E. coli*.**

| Common names of genes | Description | VITHBRA001 | VITHBRA024 | *D. radiodurans* | *E. coli* |
|---|---|---|---|---|---|
| *sodA* | Mn-containing SOD | 1 | 1 | 1 | 1 |
| *sodC* | Cu/Zn-containing SOD | 1 | 1 | 2 | 1 |
| *sodF* | Fe-containing SOD | 1 | 1 | - | 1 (*sodB*) |

**Table 9. Comparative list of peroxidases present in candidate strains, *D. radiodurans* and *E. coli*.**

| Common names of genes | Description | VITHBRA001 | VITHBRA024 | *D. radiodurans* common genes | *E. coli* |
|---|---|---|---|---|---|
| *bcp* | bacterioferritin comigratory protein | 1 | 1 | 3 | 1 |
| *tpx* | thiol peroxidase | 1 | 1 | - | 1 |
| *ahpC* | alkyl hydroperoxide reductase subunit AhpC | 2 | 2 | - | 1 |
| *ahpF* | alkyl hydroperoxide reductase subunit AhpF | 1 | 1 | - | 1 |
| *ahpE* | atypical type of AhpC | - | - | 1 | - |
| *ahpD* | alkyl hydroperoxidase D-like protein | - | - | 1 | - |
| *ohrA* | organic hydroperoxide reductase | 2 | 1 | 1 (*ohr*) | - |
| *ohrB* | organic hydroperoxide reductase | - | 1 | - | - |
| *osmC* | organic hydroperoxide reductase | - | - | 1 | 1 |
| *ccp* | cytochrome c peroxidase | - | - | 1 | - |

The candidate strains of the present study also have organic hydroperoxide resistance genes: two copies of *ohrA* in VITHBRA001 and one copy each of *ohrA* and *ohrB* in VITH-BRA024. The two copies of OhrA in VITHBRA001 share 67.38% identity to each other and 65.25% (VITHBRA001_03609) and 69.50% (VITHBRA001_04076) identity to OhrA of *B. subtilis* as compared to 53.68% and 51.47% identity with OhrB of *B. subtilis* respectively, suggesting the genes to be OhrA homologs. Ohr family of proteins are specific to the detoxification of organic peroxides; while OhrA is expressed in log phase OhrB is expressed in stationary phase [114]. Arguably, presence of two OhrA in VITHBRA001 may have supported its better resistance in vegetative state. *D. radiodurans* has one copy of Ohr and one copy of OsmC (Ohr belongs to the family of OsmC) wherein Ohr is reported to have elevated expression in post irradiation recovery phase [115] though OsmC deleted *D. radiodurans* cell showed less resistance to gamma radiation at 10 kGy [116] which suggests its importance at very high radiation exposure. In case of *E. coli*, it is observed that only a copy of OsmC is available to fight organic hydroperoxide stress.

**Bacillithiol (BSH).** BSH, similar to GSH (glutathione) found in *E.coli*, plays an important role in detoxification of reactive oxygen and protects the cysteine (thiol) containing proteins from being over-oxidized by external oxidizing agents by binding to these proteins, a condition called S-bacillithiolation [117]. WGS analysis of the candidate strains revealed the presence of four genes namely *bshA*, *bshB* and *bshC* and a *bshB* homolog which could play central role in BSH biosynthesis [118] (Table 10). VITHBRA001_04540 and VITHBRA024_02813 are seen to be homologous to BshB1 of *B. subtilis* with > 61% identity while having 64.38% identity with each other, and VITHBRA001_04681 and VITHBRA024_01391, are seen to be homologous to BshB2 of *B. subtilis* with >70% identity while having 70.14% identity with each other (S12 Fig). BshB plays a key role in deacetylase activity in the second step of BSH biosynthesis. Gaballa *et al.* (2010) reported that the two homologs of Bsh-BBS have deacetylase activity, sufficient to promote BSH production [119]. Presence of homologs for producing BSH is also reported in *Deinococcus sp.* [29], although the detailed function is yet to be worked out. Nevertheless, elevated production of BSH is reported to be observed during recovery period in *D. radiodurans* after IR irradiation [88]. VITHBRA001 encodes a thioredoxin reductase protein (TrxB- VITHBRA001_04484) which is similar to YpdA bacillithiol reductase protein (as observed in NCBI CD search) (S13 Fig) and might be required to recover from the oxidized state (BSSB) of the bacillithiol; VITHBRA024, however, does not have this gene. The function of YpdA in VITHBRA024 might apparently be carried out by other thioredoxin reductases. Additionally, both the candidate strains also possess bacilliredoxins (Brx) which could cause protein debacillithiolation by removing BSH attached to the thiol containing proteins and

**Table 10. Comparatrive list of bacillithiol biosynthesis and bacilliredoxin genes in candidate strains, *D. radiodurans* and *E. coli*.**

| Common names of genes | Description | VITHBRA001 | VITHBRA024 | *D. radiodurans* | *E. coli* |
|---|---|---|---|---|---|
| *bshA* | BSH biosynthesis glycosyltransferase | 1 | 1 | 1 | *E. coli* has glutathione (GSH) biosynthesis genes *gshA*, and *gshB*, *gor* (GSH reductase), 3 *gst* genes for GSH transferase and 4 glutaredoxin genes (*grxA-D*) |
| *bshB* | BSH biosynthesis deacetylase | 2 | 2 | 1 | |
| *bshC* | BSH biosynthesis cysteine-adding enzyme | 1 | 1 | 1 | |
| *ypdA* | BSH reductase | 1 putative YpdA bacillithiol disulfide reductase | - | 1 | |
| *brxA* | Bacilliredoxin | 1 | 1 | - | |
| *brxB* | Bacilliredoxin | 1 | 1 | - | |
| *brxC* | Bacilliredoxin | 1 | 1 | 1 (*abxC*) | |

themselves forming Brx-SSB by linking with BSH. The Brx might get reduced to its original state by BSH or by other Brx. Both the candidate strains have BrxA, BrxB and BrxC proteins (Table 10), wherein BrxA and BrxB are paralogs having CGC motif and might have similar functions. BrxC, on the other hand, possesses TCPIS motif which might have a different function as has been reported in proteins with comparable structure studied in *B. subtilis* [120, 121]. *D. radiodurans* also is reported to possess BrxC like protein named as AbxC which is reported to contribute to the oxidative stress response of *D. radiodurans* by interacting with OxyR$_{Dra}$; significantly, deletion of AbxC made *D. radiodurans* sensitive to $H_2O_2$ exposure [122].

**Thioredoxin system.** Thiols within the protein get oxidized to disulphides (in the event of oxidative stress) which are being reinstated to their reduced state with thioredoxins. Thioredoxin reductases help reduce the oxidized thioredoxin so as to get back to their reduced state. Subsequently, the oxidized thioredoxin reductase, gets back to the reduced stage by accepting electron from NADPH [123]. WGS analysis reveals that VITHBRA001 and VITHBRA024 possess one homolog each of thioredoxin 1 (*trxA*) gene and two *trx*-like genes. Multiple sequence alignment (MSA) confirms the presence of TrxA in both the candidate strains, evidenced by the presence of the conserved WCGPC motif and the absence of CXXC motif which is exclusive to TrxC. TrxA of candidate strains are more similar to TrxA$_{Dra}$ than to TrxA$_{Eco}$ with >50% similarity. It is reported that deletion of TrxA as well as TrxC in *D. radiodurans* has resulted in decreased survivability for cells exposed to $H_2O_2$ [124] whereas in *E. coli* deletion of TrxC and not of TrxA makes it sensitive to $H_2O_2$ [125]. Trx-like genes of the candidate strains have WCPDC motif comparable with that of DR2085 gene of *D. radiodurans* (S14 Fig). The exact function of this gene is still elusive in *D. radiodurans* [126], although irradiation is reported to have induced its expression [127].

VITHBRA001 is observed to possess three copies of thioredoxin reductase (*trxB*) while VITHBRA024 has only one copy. TrxB protein, VITHBRA001_04258 and VITHBRA024_01610 show 45.71% and 45.40% identity respectively, with TrxR protein of *D. radiodurans* (S15 Fig) and ~38% identity with *E. coli* (S15 Fig), while the other two homologs of TrxB of VITHBRA001, such as VITHBRA001_01782 and VITHBRA001_04043, although confirmed to have TrxB domain through NCBI CD search analysis (Fig 9), show identity < 23% (Table 11 and S15 Fig). The thioredoxin systems also help recover the antioxidants such as Bcp and AhpC to their reduced state. Presence of additional *trxB*-like genes in VITHBRA001 is presumed to give added support in resolving disulphides formed by ROS. The relevance of thioredoxin reductase gene (*trxR*) in survival of the cell against $H_2O_2$ exposure has been

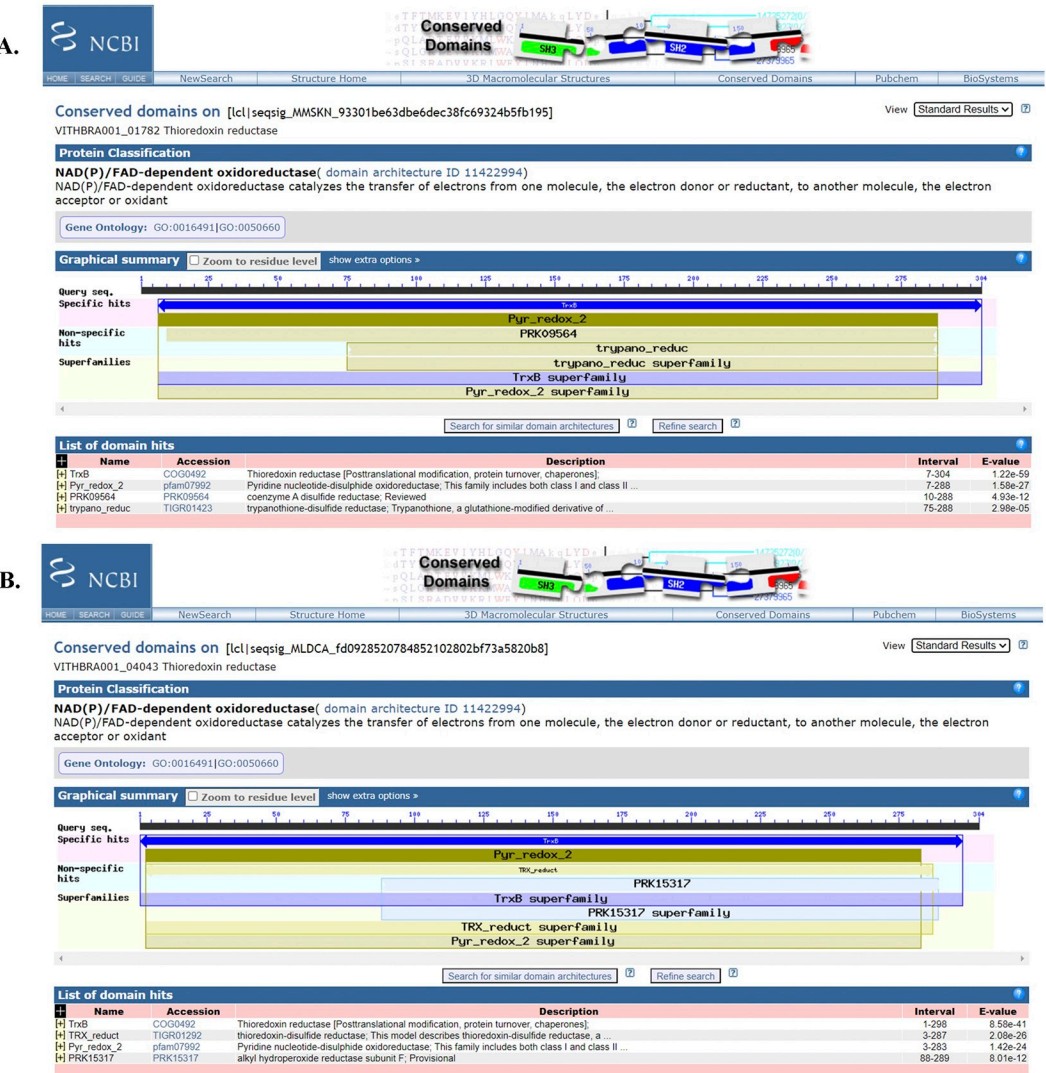

**Fig 9. NCBI CD search analysis of proteins VITHBRA001_01782 and VITHBRA001_04043.** Figures (A) and (B) are the NCBI CD analysis for proteins VITHBRA001_01782 and VITHBRA001_04043 respectively which show specific hits with TrxB domain confirming them to be thioredoxin reductases.

proven in *D. radiodurans* inasmuch as its silencing causes the cell to become sensitive to $H_2O_2$ [128].

Further, the role of Trx system in unison with MsrA, MsrB and MsrC in the conversion of methionine sulfoxide to methionine (resultant of oxidation due to ROS) has been demonstrated by previous research. While MsrA works on methionine S-sulfoxide, MsrB and MsrC would act on Methionine R-sulfoxide and free oxidized methionine, respectively [129]. VITHBRA024 has all the three genes of Msr while VITHBRA001 has a fused gene with both the functional domains of MsrA and MsrB (N-terminal and C-terminal respectively) and a separate MsrC (Table 11, Fig 10). A fused MsrAB is reported to be more efficient enzymatically than separate proteins having the same function [130, 131]. The *msr* genes are also present in *E. coli* and *D. radiodurans*. Its functional importance is reiterated by MsrA expression in recovery phase of *D. radiodurans* after 3 kGy dose of irradiation [132] and MsrB expression in recovery period after exposure to high levels of gamma radiation [133].

**Table 11. Comparative list of genes involved in thioredoxin system and recycled by this system in candidate strains, *D. radiodurans* and *E. coli*.**

| Common names of genes | Description | VITHBRA001 | VITHBRA024 | *D. radiodurans* common genes | *E. coil* |
|---|---|---|---|---|---|
| *trxA* | Thioredoxin | 1 | 1 | 1 | 1 |
| *trxA-like* | Thioredoxin | 2 | 2 | - | - |
| *trxB* | Thioredoxin reductase | 1 | 1 | 1 | 1 |
| *trxB-like* | Thioredoxin reductase homologs | 2 | - | - | - |
| *trxC* | Thioredoxin 2 | - | - | 1 | 1 |
| *msrA* | peptide methionine sulfoxide reductase MsrA | - | 1 | 1 | 1 |
| *msrB* | peptide methionine sulfoxide reductase MsrB | MsrA and MsrB is fused in one protein. | 1 | 1 | 1 |
| *msrC* | putative methionine R-sulfoxide reductase | 1 | 1 | - | 1 |
| *frnE* | DsbA-like cytoplasmic disulphide oxidoreductase | 1 | - | 1 | - |

VITHBRA001 is also observed to have a homolog of *frnE* gene with 43.15% identity with a novel DsbA-like cytoplasmic disulphide oxidoreductase found in *D. radiodurans* (DR0659) and has the canonical CPFC motif instead of CPWC motif of *D. radiodurans* (Fig 11); significantly, CPFC motif is observed in *D. proteolyticus* and *D. maricopensis*. Alongside this, it also has the CxxxxC motif near to its C-terminal end as reported in *D. radiodurans* and other *Deinococcus* sp. (Fig 12). Its mutation in *D. radiodurans* is reported to make the cell IR and $H_2O_2$ sensitive [29, 134].

**Mn/Fe homeostasis.** High $Mn^{2+}$ as compared to $Fe^{2+}$ is shown to be essential to stop ROS from aggravating in the system by reducing Fenton's reaction. Mn forms complexes with phosphates, nucleosides and amino acids within the cell to form non-enzymatic metabolites which immensely contribute to the ROS quenching and protect oxidation of proteins within the cell [135]. Mn-phosphate and Mn-carbonates are reported to be very efficient in removing ROS from the cell and in particular the superoxides [136]. Radiation resistant bacteria are known to maintain high Mn/Fe Ratio with the help of multiple transport and regulatory proteins [100]. VITHBRA001 seems to rely on ABC transporter system for Mn/Fe homeostasis, judged by the WGS data, revealing the presence of consecutively arranged four genes (*mntA*, *mntB*, *mntC* and *mntD*) which are suggested to play central role in Mn uptake [137, 138]. VITHBRA024, on the other hand, encodes genes for both ABC-type complex and NRAMP transporter systems evidenced by the occurrence of *mntA*, *mntB* and mntC/ *mntD* gene in an operon along

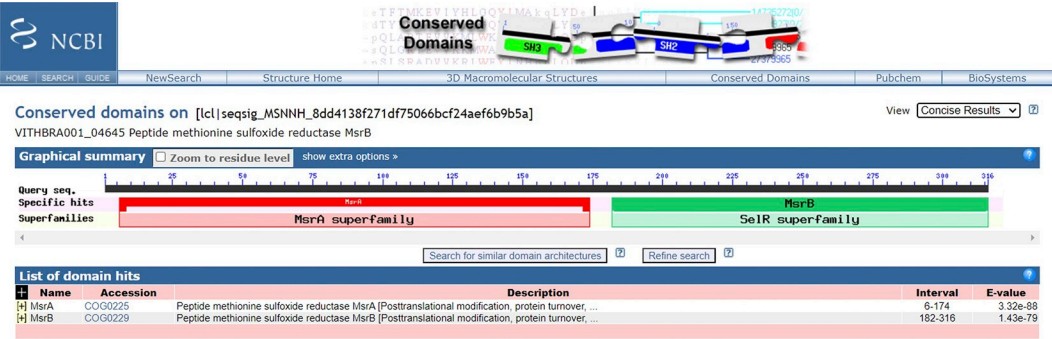

**Fig 10. NCBI CD search analysis of VITHBRA001_04645.** The analysis shows the presence of two functional domains of MsrA and MsrB fused in one protein of VITHBRA001_04645.

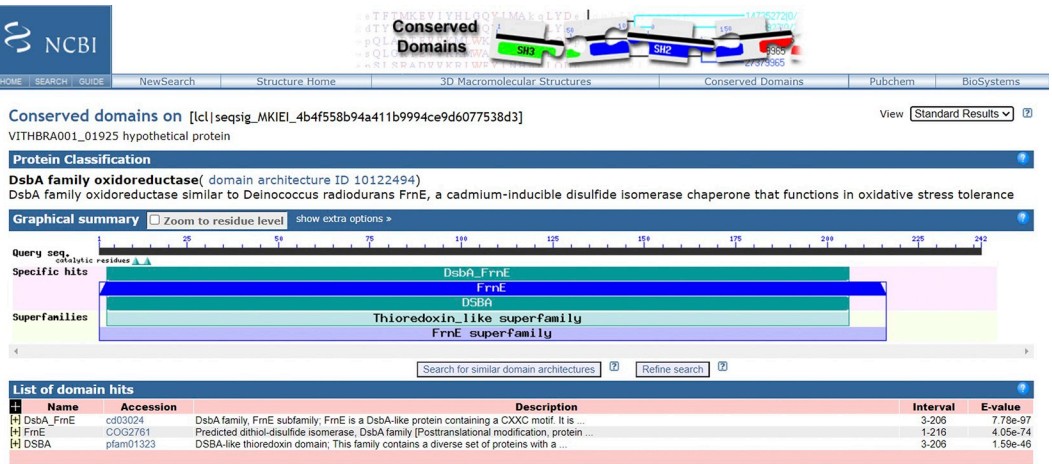

**Fig 11. NCBI CD search analysis for VITHBRA001_01925.** The figure confirms that VITHBRA001_01925 is a DsbA family protein which is similar to the *D. radiodurans* FrnE novel cytoplasmic disulphide oxidoreductase protein.

with the NRAMP transporter *mntH*, a situation on par with *D. radiodurans* and few other *Deinococcus* sp. (Table 12). MSA of all the *mnt* genes of both the strains suggests that MntC in the ABC complex of VITHBRA024 share more identity with MntD than with MntC of VITHBRA001 (S16 Fig). NCBI CD search analysis, however, suggests functional similarity for *MntC* and *MntD* of VITHBRA001; NCBI CD search further revealed the presence of an extended C-terminal region in *MntC*, the relevance of which is unclear. Most of the *E. coli* strains possess an MntH while a few pathogenic strains are reported to possess *sitABCD* transporter which can uptake both Mn and Fe based on the requirement of the cell and is not specific to Mn

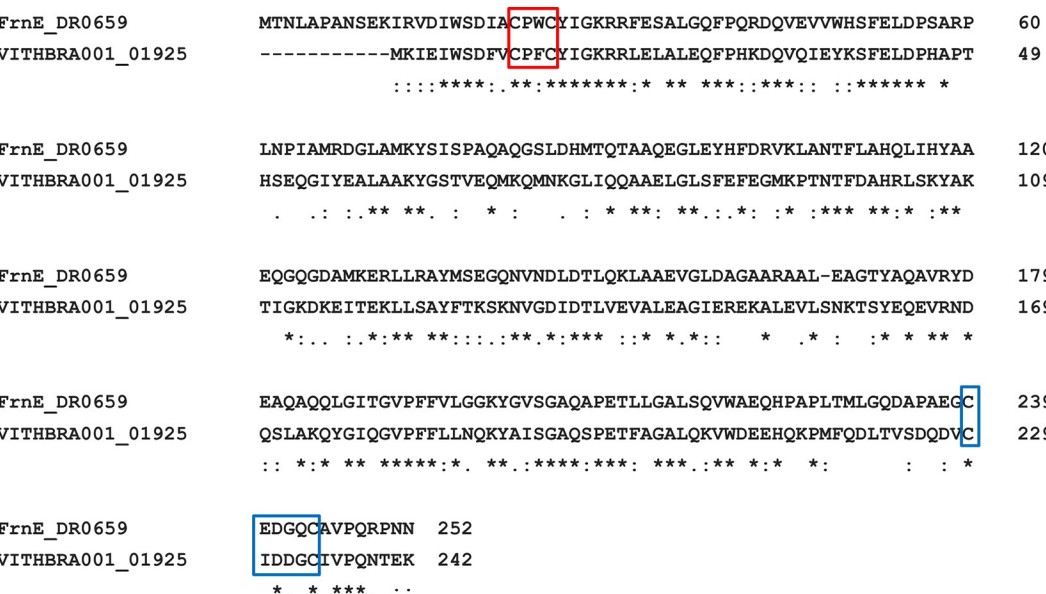

**Fig 12. Pairwise alignment of FrnE protein from *D. radiodurans* and VITHBRA001.** The red box in the figure highlights the highly conserved active site CXXC motif of FrnE protein (a cytoplasmic thiol oxidoreductase protein) while the blue box highlights the CxxxxC motif which is present in *Deinococcus* species except for *D. maricopensis* and might help in regeneration of the active site. The alignment was done using Clustal omega version 1.2.2.

**Table 12. Comparative list of genes responsible for manganese uptake and availability of phosphate to produce non-enzymatic metabolites in the candidate strains, *D. radiodurans* and *E. coli*.**

| Common names of genes | Description | VITHBRA001 | VITHBRA024 | *D. radiodurans* | *E. coli* |
|---|---|---|---|---|---|
| *mntH* | Mn$^{2+}$ transporters of the NRAMP family | - | 1 | 1 | 1 |
| *mntA* | ABC-type Mn$^{2+}$ transport system, Mn$^{2+}$-binding component | 1 | 1 | 1 | - |
| *mntB* | ABC-type Mn$^{2+}$ transport system, ATPase component | 1 | 1 | 1 | - |
| *mntC* | ABC-type Mn$^{2+}$ transport system, permease component | 1 | - | 1 | - |
| *mntD* | ABC-type Mn$^{2+}$ transport system, permease component | 1 | 1 | - | - |
| *ppk1* | polyphosphate kinase 1 | 1 | - | 1 | 1 |
| *ppx* (COG0855) | exopolyphosphatase | 1 | - | 1 | 1 |
| *ppk2* | polyphosphate kinase 2 | - | - | 1 | - |

[139, 140]. Significantly, among the two systems, ABC type seems to be more prevalent (than NRAMP) in Mn uptake, due to its ubiquitous presence in all the radiation resistant *Deinococcus* sp. examined so far [29]. Balancing of Mn/Fe ratio is also effected by maintaining iron homeostasis wherein Dps-like proteins play a crucial role [141]. These are ferritin family proteins responsible for iron sequestration, ferroxidation and binding to DNA to protect it from oxidative damage [100, 141]. Bourne out of the genome analysis, VITHBRA024 has two such proteins i.e., Dps and MrgA, while VITHBRA001 has one Dps. Further COG analysis using RPS-BLAST brings out three more Dps–like proteins (ascertained by Uniport–BLAST and NCBI CD search) arranged sequentially in the genome (VITHBRA001_00185, VITH-BRA001_00186 and VITHBRA001_00187) (Fig 13). Significantly, in *E. coli* one *dps* gene is present which provides protection under oxidative stress [142] and in *D. radiodurans*, two *dps* genes are present of which Dps-2 is suggested to protect against exogenously derived reactive oxygen species [143].

Searching through the Prokka annotation file, VITHBRA001 is observed to have a copy of Ppk1 (VITHBRA001_3565) which has 43.03% identity with the Ppk1 of *D. radiodurans* (DR1939), and a copy of exopolyphosphatase (Ppx) (VITHBRA001_3564) which has an identity of 23.87% with the exopolyphosphatase (DRA0185) of *D. radiodurans* (Table 12). Under oxidative stress, the quantity of polyphosphate increases in the cell with the help of polyphosphate kinase 1 (Ppk1) which then acts as chaperone to protect proteins from mis-folding [144]. This polyphosphate has also been reported to chelate Mn$^{2+}$ and store with them [145]. The Ppx is responsible to cleave the phosphoanhydride bond of polyphosphates [146] and produce phosphates which can then bind to Mn$^{2+}$ to produce the small non-enzymatic metabolites. *D. radiodurans* not only contains Ppk1 and Ppx but also another kinase Ppk2 which is also responsible for hydrolysis of polyphosphate [29, 147, 148]. Our finding shows that these essential genes for IR resistance are not present in VITHBRA024. Though VITHBRA024 strain consists of *mntH* gene that is more efficient in Mn$^{2+}$ uptake, but lack of Ppk1 and Ppx may not provide the cell with the much-needed phosphate molecules to bind with Mn$^{2+}$ for ROS removal, which can be a major factor for its less resistance capacity as compared to VITHBRA001. This hypothesis is substantiated by Dai *et al.* (2021), wherein *ppx* gene when disrupted in *D. radiodurans* showed increased sensitivity to oxidative stress [149].

**DNA damage response (DDR).** The major system of DNA damage response in bacteria is the SOS pathway wherein the RecA nucleofilament formation leads to the self-cleavage of LexA dimer which otherwise in its dimeric state represses the SOS-regulon genes [150]. Borne out from WGS analysis, the presence of *recA* and *lexA* genes suggests the presence of SOS-pathway. Significantly, each of the candidate species has two homologs of the LexA, out of which VITHBRA001_03904 and VITHBRA024_01120, show more than 83% and 92% identity

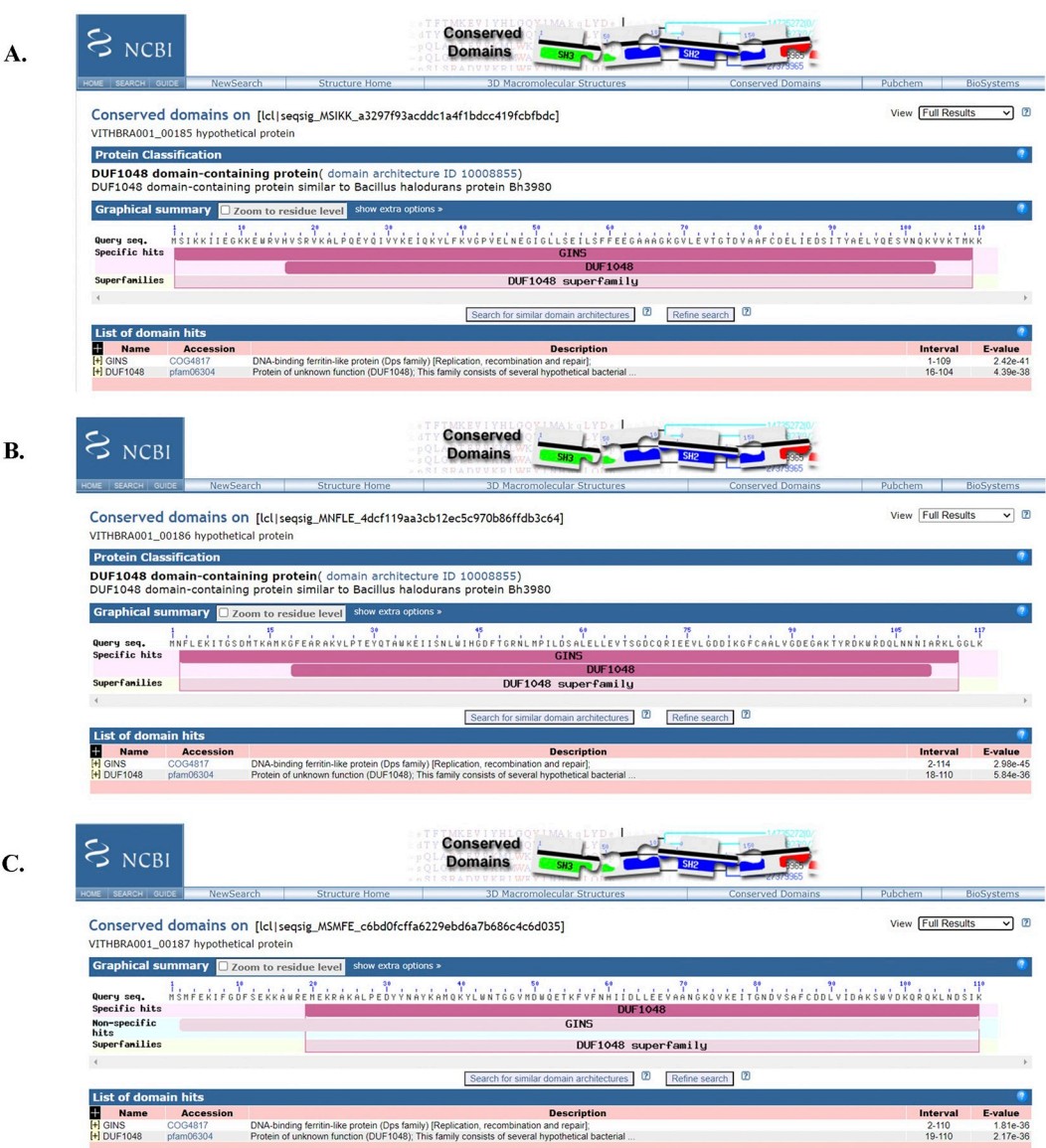

**Fig 13. Dps-like proteins in VITHBRA001.** Three consecutive genes/proteins were found to belong to Dps family of proteins having ferritin like function. NCBI CD search analysis confirms this function for all three proteins represented below as (A), (B) and (C). This observation was preliminarily done through COG analysis.

respectively with LexA of *B. subtilis*, suggesting functional similarity. VITHBRA024_00024, the other LexA homolog with 48.57% identity with LexA of *B. subtilis*, however, is a short sequence having only the N-terminal DNA binding domain of LexA (judged from CD analysis); hence its functionality is speculative. The other homolog of LexA, VITHBRA001 (00738), is a complete sequence, but has only 19.35% identity with VITHBRA001_03904 and 16.67% identity with LexA of *B. subtilis*, however, it has 23.50% identity with one out of two LexA of *D. radiodurans* (DR_A0344). It also deserves mention that the DNA repair genes may not always be under the control of SOS pathway, and can be under the control of other SOS-independent pathways such as DdrO/IrrE in *D. radiodurans*, PafBC in *Mycobacterium sp.* and DriD in *Caulobacter crescentus* [151, 152].

**Transcription regulators.** Presence of *perR* and absence of *oxyR* (present in *E. coli*) in both the candidate strains (judged from WGS analysis), would indicate their dependence on PerR regulon (as observed in most Gram-positive bacteria) for defence against ROS. VITH-BRA001 is found to have one copy of *perR* gene (VITHBRA001_02079), while VITHBRA024 has three homologs of *perR* (VITHBRA024_03988 as PerR1, VITHBRA024_01028 as PerR2 and VITHBRA024_02004 as PerR3); the nomenclature has been chosen, based on its identity with *Bacillus licheniformis* [153]. PerR is a mettaloregulator which in its $Fe^{2+}$ bound state, represses its regulatory downstream proteins and derepresses them on interaction with $H_2O_2$, causing oxidation of its histidine residue with production of hydroxyl ion and $Fe^{3+}$ [154]. Further, PerR is a regulon that controls several genes involved in peroxide removal (*ahpC*, *ahpF* and *katA*), sequestering of iron (*mgrA*), iron homeostasis regulation (*fur*, through its capability to regulate iron transporters), iron efflux (*zosA*, renamed as *pfeT* [155]) and heme biosynthesis to support heme binding catalases (*hemAXCDBL*) [156]. It could be reasonably presumed that VITHBRA001, in spite of the absence of *mgrA* gene, could use its PerR regulon to perform all the functions related to iron sequestration; the presence of Dps (a homolog of MgrA) and Dps-like protein, could offset that deficiency in VITHBRA001.

The three PerR homologs of VITHBRA024 (*Bacillus paralicheniformis*) (PerR1, PerR2 and PerR3) are quite comparable with PerR$_{Bli}$, PerR2 and PerR3 of its closest phylogenetic ally *Bacillus licheniformis* (Bli), with identities of 100%, 99.3% and 97.1% respectively, suggesting functional similarity. In *B. licheniformis*, both PerR$_{Bli}$ and PerR2 have repressor activity and may influence the PerR regulon either simultaneously or may affect individually based on the extent of severity of oxidative stress on the cell [153]. The PerR3 of VITHBRA024 seems to have a strategy on par with that of *B. licheniformis*, to overcome $H_2O_2$ stress, judged by the percentage of identity. In VITHBRA024, as in *B. licheniformis*, PerR3 is situated in between *katA* gene and putative ferro-chetalase gene. In *B. licheniformis*, this set of three genes get co-expressed when under $H_2O_2$ stress [157]; it could be probable that the same mechanism (of co-expression) is the rule in VITH-BRA024 also. PerR3, however, has no repressor function like the other two *perR* genes [153].

The presence of MntR regulator within the candidate strains and absence of *mur* gene suggest that the homeostasis of manganese ion could be regulated by MntR, by repressing the Mn transporters, so as to save the cell from Mn toxicity, once the cell attains Mn sufficiency [158]. In addition to *mntR* like gene, *D. radiodurans* possesses *mur* gene that probably helps upregulate $Mn^{2+}$ transporters [159]. That MntR is important in maintaining Mn homeostasis in the candidate strains can be suggested by the study of mutant MntR of *B. subtilis*, wherein its mutation caused the cell to have less ROS as derepression of Mn transporters led to high Mn acquisition which has antioxidant properties [160].

Organic peroxides produced in cell due to ROS could be sensed by *ohrR* gene present in one copy within both the strains under study. This *ohrR* could be responsible to activate *ohrA* gene that plays a crucial role in hydroperoxide resistance [154]. Some of the regulators which sense oxidative stress in various organisms include *spx* gene. VITHBRA001 encodes two *spx* genes (VITHBRA001_00957 and VITHBRA001_03158) having 68.7% identity with each other and VITHBRA024 (VITHBRA024_00364) encodes one. In addition to *spx*, a homolog of this gene, *mgsR*, is also present in one copy with both the strains (VITHBRA001_00837 and VITH-BRA024_02538). Both the homologs have similar function but they are activated under the influence of $\sigma^A$ and $\sigma^B$ respectively (present within the genome of the strains under study) and again in the exponential growth phase and stationary phase respectively [161]. Spx regulates transcription of several genes under the effect of oxidative stress and is particularly influenced by the formation of disulfide in several thiol containing proteins. Thus, to combat oxidation of such proteins within the candidate strains, the genes such as *trxA*, *trxB*, *msrA* and *msrB* along with bacillithiol would be upregulated by Spx [162]. Many bacteria are reported to have

multiple Spx and apparently, based on various environmental factors different Spx regulates different stressors [163]. Thus, presence of two *spx* genes in VITHBRA001 could be presumed to help better oxidative stress management.

**Pigments as antioxidants.** Pigments are known to afford major protective mechanism against radiation among microbes of various domains, and have significant antioxidant properties [164, 165]. Further, pigments are known to be act as antioxidants, and help remove ROS from organisms [166]. VITHBRA001 produces a pigment of orange hue. Up on being inspected for the natural products being synthesized by VITHBRA001 using antiSMASH and also using KEGG database, a cluster of genes responsible for $C_{30}$ carotenoid biosynthesis became discernible. $C_{30}$ carotenoids are suggested to have similar or more antioxidant properties than $C_{40}$ carotenoids (widely found in the environment) [167]. Borne out from annotation softwares, VITHBRA001 is observed to have five pigment synthesizing genes in cluster namely *crtO*, *crtQ*, *crtP*, *crtM*, and *crtN*. Further, antiSMASH tool could detect four genes (*crtQ*, *crtP*, *crtM*, *crtN*), having similarity score of 0.61 with the four carotenoid producing genes of *Halobacillus halophilus* (*orf-GT*, *crtNb*, *crtM*, *crtNa* respectively) and 0.75 similarity score with two genes from *Rhodobacter sphaeroids* (*crtB* and *crtI*). The standalone BLAST of the protein FASTA file of VITHBRA001 with the well-defined $C_{30}$ carotenoid genes of *Staphylococcus aureus*, *H. halophilus* and two other reported carotene producing species of *Metabacillus* (*M. flavus* and *M. indicus*), revealed that CrtQ, CrtN, and CrtP have higher identity with respective proteins of *S. aureus*, whereas CrtM (of VITHBRA01) was found more identical to that of CrtM of *H. halophilus*. There were no hits of CrtO with any organism other than with CrtO of *S. aureus*. Additionally, another *crtP* gene is present in VITHBRA001 on another location of the genome and was not detected by antiSMASH, but detected through standalone BLAST. The second copy of CrtP also shows similarity with the CrtNa, CrtNb and CrtNc of all selected organisms but the highest identity is with CrtP of *S. aureus* (S1 Table). Thus, we can suggest that the carotenoid product of the candidate strain is similar to the staphyloxanthin of *S. aureus*. It is reported that *S.aureus* with disrupted staphyloxanthin biosynthesis gene (Δ*crtM*) was more sensitive to ROS as compared to its wild type [168].

## Comparison of the candidate strains with *D. radiodurans* and *E. coli*–a consolidated genomic approach

Commonality of the candidate strains with *D. radiodurans* exists in the context of ROS removal such as the catalases—*katX* and *katE*, superoxide dismutase–*sodA* and *sodC* and peroxidases like *bcp*, and protection of protein from oxidative damage through *trxA*, *trxB*, *msrA*, *msrB* and bacillithiol along with bacilliredoxin (Table 13). *E. coli* also shares some of the above genes but differences include in the number of peroxide scavenging genes. *E. coli* has fewer genes such as two catalases, one *ahpC*, and one *osmC* (Tables 7, 9 and 11). Presence of different peroxidases in *D. radiodurans* and the candidate strains may help in better resistance. Further, presence of MnKat in candidate strains could be an advantage in resistance compared to the situation in *E. coli*. Cleaning of damaged protein from cells through the activation of proteases, Lon and ClpXP, are also observed in candidate strains and disruption of these proteins from *D. radiodurans* makes cells sensitive to radiation [4]. *E. coli* also possesses these genes but it is reported that genes such as *clpX* has positive Darwinian selection in IRRB for better adaptation under IR stress, which is not observed in *E. coli* [169]. Presence of Mn uptake genes (*mntABCD* and *mntH*) show the importance of maintaining the Mn/Fe homeostasis within the cell which ultimately helps the cell to resist radiation.

Both the candidate strains also display presence of orthologous genes that help DNA repair akin to the situation in *D. radiodurans* (Table 13); for instance, complete RecFOR pathway for

**Table 13. List of consolidated proteins between candidate strains and *D. radiodurans*.**

| Sl no. | VITHBRA001 + VITHBRA024 + Dra | Only for VITHBRA001 + VITHBRA024 | VITHBRA001 + Dra | VITHBRA024 + Dra | VITHBRA001 exclusive | VITHBRA024 exclusive | Dra exclusive |
|---|---|---|---|---|---|---|---|
| 1 | RecA [HR] | AddA [HR] | RuvC [HR] | MsrA [TH] | UvsE [NER] (VITHBRA001_01817) | RecS [HR, Rep] | UvrD [HR, NER] |
| 2 | RecF [HR] | AddB [HR] | UvrA2 [NER] | MsrB [TH] | MutM [BER] (VITHBRA001_02212) | PerR2 [TR] | Udg4 [BER] |
| 3 | RecO [HR] | Ku [NHEJ] | UvsE [NER] | MntH [MH] | Nfo [BER] (VITHBRA001_02031) | PerR3 [TR] | Nfi [BER] |
| 4 | RecR [HR] | LigD [NHEJ] | Mug [BER] | | MutS2 [Oth] (VITHBRA001_02893) | OhrB [PX] | Nth (2) [BER] |
| 5 | RecJ [HR] | PcrA [HR, NER] | XthA [BER] | | Spx [TR] (VITHBRA001_00957) | MrgA [ISeq] | Top1A [OR] |
| 6 | RecN [HR] | YjcD [Oth] | YpdA [BSH] | | MnKat [ROR] (VITHBRA001_04223) | | YqgF [OR] |
| 7 | RecQ [HR] | Nfo [BER] | MntC [MH] | | OhrA [PX] (VITHBRA001_04076) | | NERD domain [OR] |
| 8 | RecD2 [HR] | PolC [HR, Rep] | Ppk1 [Cap, ROR] | | TrxB (2) [TH] | | HepA [OR] |
| 9 | RuvA [HR] | DinB [TEL] | Ppx [ROR] | | Dps-like (3) [ISeq] | | PARG [OR] |
| 10 | RuvB [HR] | Fur [TR] | FrnE [TH] | | MsrA-MsrB [TH] | | PNKP [OR] |
| 11 | RecG [HR] | Zur [TR] | MsrP [TH] | | | | Rnl [OR] |
| 12 | RecX [HR] | Spx [TR] | | | | | RqkA [OR] |
| 13 | RadA [HR] | MgsR [TR] | | | | | OxyR1 [TR] |
| 14 | SbcC [OR] | MntR [TR] | | | | | OxyR2 [TR] |
| 15 | SbcD [OR] | OhrR [TR] | | | | | Mur [TR] |
| 16 | RarA [HR] | SigA [TR] | | | | | Irr [TR] |
| 17 | UvrA1 [NER] | SigB [TR] | | | | | DtxR [TR] |
| 18 | UvrB [BER] | KatA [ROR] | | | | | DdrA [SDSA] |
| 19 | UvrC [BER] | MnKat [ROR] | | | | | DdrB [SDSA] |
| 20 | AlkA [BER] | SodF [ROR] | | | | | DdrC [Oth] |
| 21 | Ung [BER] | Tpx [PX] | | | | | DdrD [Oth] |
| 22 | MutY [Nud] | 2-cys AhpC [PX] | | | | | DdrE [Oth] |
| 23 | MutM (Fpg) [BER] | 3-cys AhpC [PX] | | | | | DdrF [Oth] |
| 24 | Nth [BER] | AhpF [PX] | | | | | DdrG [Oth] |
| 25 | MutL [MMR] | Trx-like [TH] | | | | | DdrH [Oth] |
| 26 | MutS [MMR] | OhrA [PX] | | | | | DdrI [TR] |
| 27 | MutS2 [HR] | MsrC [TH] | | | | | DdrJ [Oth] |
| 28 | HelD [OR] | BshB2 [BSH] | | | | | DdrK [Oth] |
| 29 | LigA [HR, BER] | BrxA [BSH] | | | | | DdrL [Oth] |
| 30 | PolA [Rep, BER, MMR] | BrxB [BSH] | | | | | DdrM [Oth] |
| 31 | PolX [Rep] | MntD [MH] | | | | | DdrN [Oth] |
| 32 | DnaE [Rep] | | | | | | DdrO [TR] |
| 33 | DnaN [Rep] | | | | | | DdrP [Oth] |
| 34 | DnaX [Rep] | | | | | | PprA [Oth] |
| 35 | PriA [Rep] | | | | | | IrrE [TR] |
| 36 | DnaA [Rep] | | | | | | PprM [Oth] |
| 37 | SsbA [HR, Rep] | | | | | | SodC [ROR] |
| 38 | GyrA [Rep] | | | | | | Bcp (2) [PX] |
| 39 | GyrB [Rep] | | | | | | DR2085 [TH] |
| 40 | TopA [Rep] | | | | | | AhpE [PX] |
| 41 | XseA [OR] | | | | | | AhpD [PX] |
| 42 | XseB [OR] | | | | | | TrxC [TH] |

*(Continued)*

**Table 13.** (Continued)

| Sl no. | VITHBRA001 + VITHBRA024 + Dra | Only for VITHBRA001 + VITHBRA024 | VITHBRA001 + Dra | VITHBRA024 + Dra | VITHBRA001 exclusive | VITHBRA024 exclusive | Dra exclusive |
|---|---|---|---|---|---|---|---|
| 43 | Pnp [HR, NHEJ, Ox] | | | | | | CCP [PX] |
| 44 | LexA-ArsR [TR] | | | | | | OsmC [PX] |
| 45 | LexA-XRE [TR] | | | | | | Ohr [PX] |
| 46 | PerR [TR] | | | | | | YhfA [PX] |
| 47 | KatX [ROR] | | | | | | Dps2 [ISeq] |
| 48 | KatE [ROR] | | | | | | Ppk2 [ROR] |
| 49 | SodA [ROR] | | | | | | DsbA-like protein [TH] |
| 50 | SodC [ROR] | | | | | | PqqE [Ox] |
| 51 | Bcp [PX] | | | | | | MsrQ [TH] |
| 52 | TrxA [TH] | | | | | | DR_B0067 [Ox] |
| 53 | TrxB/TrxR [TH] | | | | | | |
| 54 | BshA [BSH] | | | | | | |
| 55 | BshB1 [BSH] | | | | | | |
| 56 | BshC [BSH] | | | | | | |
| 57 | BrxC/AbxC [BSH] | | | | | | |
| 58 | MntA [MH] | | | | | | |
| 59 | MntB [MH] | | | | | | |
| 60 | Dps [ISeq] | | | | | | |
| 61 | Lon1 [PR] | | | | | | |
| 62 | Lon2 [PR] | | | | | | |
| 63 | ClpX [PR] | | | | | | |
| 64 | ClpP [PR] | | | | | | |
| 65 | Hsp33 [Cap] | | | | | | |

Dra = *D. radiodurans*, HR = Homologous recombination, NHEJ = Non-homologous end joining, SDSA = Synthesis-dependent strand annealing, Rep = DNA replication related genes which also involved in repair, NER = nucleotide excision repair, BER = Base excision repair, Tel = Translesional repair, ISeq = Iron sequestration, ROR = Removal of ROS, PX = Peroxidases, TH = Thiol reductases, BSH = Involved with Bacillithiol system, TR = Transcriptional regulators, MH = Manganese homeostasis, PR = damaged protein removal, Cap = Chaperones, Oth = proteins with hypothetical function, OR = Other repair related proteins, Ox = Other oxidative stress management proteins, Nud = Nudix. The number in brackets represents the number of homologs which are exclusive to that protein present. Locus tag in brackets has been mentioned for the proteins which are exclusive to that organism, but share the same name with its homolog, and the homolog is common to another organism in this study.

DSB repair of DNA, variety of BER glycosylases and AP lyase genes, repair of oxidatively produced DNA adducts through UvrABC pathway and repair of mismatches through the well-established MMR pathway. However, most of these genes are common to radiation sensitive *E. coli* as well (Tables 2–12) while some differences are worth noting. For instance, RecFOR pathway is present in both IRRB and *E. coli*, but in *E. coli* RecBCD pathway is also present similar to few IRRB organisms (*K. radiotolerans*, *R. erythropolis*); but it is still elusive which pathway is significantly activated. Although, a hint for preference is observed in Li *et. al.* (2015) wherein the transcriptomics of *K. radiotolerans* reports the upregulation of *recR* gene which is significantly important in RecFOR pathway [15]. Additionally, RecF, RecR and RecN were found to contribute to the enhanced resistance of *E. coli*, generated through directed evolution [170]. These reports tempt us to suggest that RecFOR pathway is preferred over RecBCD in IR resistance. *D. radiodurans* also seem to have NHEJ pathway, though not similar to the candidate strains. Genes such as *sbcC* (ATPase) and *sbcD* (nuclease) act as a complex and remove

hairpin structure from the DSB ends to facilitate the HR, ESDSA or NHEJ repair [171]. A similar function of end resection is also performed by *polX* (3′-to-5′ double-strand exonuclease), and disruption of all three genes makes *D. radiodurans* cell gamma radiation sensitive [171]. It is also reported that *polX* has a positive Darwinian selection in IRRB [169]. These three genes are also common to the candidate strains and *E. coli* lacks *polX* (S2 Table). A significant difference is also observed in the number of nudix genes, particularly MutT family of genes, present in IRRB and in *E. coli*. Through literature and percentage identity analysis in our study, it was observed that *D. radiodurans* possessed eight MutT-like genes [172], VITHBRA001 had five and VITHBRA024 had two, whereas, *E. coli* had only one (Table 6). This might give a better edge to the IRRB organisms to get rid of 8-oxo-dGTP or 8-oxo-GTP—an abundantly formed oxidized nucleotide on exposure to radiation [80, 173]. *D. radiodurans* has strikingly different genetic machinery to cope with high radiation dose. For instance, *D. radiodurans* has a battery of genes designated as *ddr* and *ppr* which are specific to the *Deinococcus* group of species [29] and are responsible for recombination repair such as SDSA, act as transcriptional regulators and many are hypothetical proteins whose function is yet to be determined. It has specialized IrrE/DdrO transcription factors which bind to a particular RDRM region upstream of many DNA repair gene and trigger DDR on being irradiated by IR [174]. However, the candidate strains as well as *E. coli* possess SOS pathway for DDR, but the difference in the rate of activation of SOS pathway can cause the difference in radiation resistance. Such a condition has been experimentally proven by Simmon *et. al.*, (2009), wherein only 5% of *B. subtilis* population activated SOS regulon while 87% of *E. coli* population was activated after exposure to IR and *B. subtilis* survived better [175]. Further, there are specific histidine kinases in *D. radiodurans* whose silencing makes it highly sensitive to IR exposure. Most of the genes that are relevant for IR and oxidative stress resistance in *D. radiodurans* are triggered through these components as well [59]. *D. radiodurans* also has multiple homologs of DNA repair genes such as *nth* and *bcp*, various functionally undefined *trx*-like genes (apparently related to thiol-reduction) and reduced iron centric proteins related to reduced Fenton reaction. Further, *D. radiodurans* reports the absence of genes responsible for translesional repair of DNA which are prone to higher error rate during DNA repair [29].

## Comparison between the candidate strains (VITHBRA001 and VITHBRA024)–a genomic approach

The authors address the striking differences existing between the two candidate species, helpful to explain the higher radiation tolerance displayed by VITHBRA001.

- The presence of *ruvC*–like gene in VITHBRA001, which is suggested to cleave cruciform junctions, structurally analogous to Holliday junctions, by introducing nicks into strands and help this organism in resolving Holliday junction during DNA repair through HR [176]. VITHBRA024, however, does not have RuvC.

- VITHBRA001 has 11 genes performing glycosylase, AP lyase and endonuclease activities which could efficiently carry out Base Excision Repair, whereas this function is being accomplished by only seven genes in VITHBRA024. The preponderance in the pool of genes for glycosylase, AP lyase and endonuclease activity would give VITHBRA001 an edge for radiation resistance over VITHBRA024. Endonucleases are important to complete BER wherein they cleave the AP sites, producing a 5′- abasic sugar flap and a 3′-OH site for the DNA Pol I and LigA to refill the gap [177]. Further, *xthA* exonuclease gene and two *nfo* endonuclease gene possessed by VITHBRA001 are reported to have important role in repairing oxidatively damaged DNA [177–179]; VITHBRA024 has only a single copy of *nfo* endonuclease gene.

- VITHBRA001 has a strong NER mechanism functional through two pathways, UvrABC pathway and UVER (UvsE) pathway. *uvsE* gene (absent in VITHBRA024) forming the UVER pathway initiates a cascade of DNA repair which is similar to that of the UvrABC pathway [180] contributing to efficient NER system.

- Possession of three homologs of *trxB* gene by VITHBRA001 (against one *trxB* homolog in VITHBRA024) would offer better oxidative stress management. Involvement of multiple Trx system in efficient oxidative stress management has been reported of late in *Clostridioides difficile* (a Firmicute) [181].

- Further, presence of fused *msrAB* gene, housed by VITHBRA001 would be catalytically more efficient than individual genes of VITHBRA024 [131].

- We could also find that VITHBRA001 (but absent in VITHBRA024) is in possession of *frnE* gene–a novel gene which has been proven to help efficient radiation resistance in *D. radiodurans*.

- Genes like *ppkI* and *ppx* present in VITHBRA001 (absent in VITHBRA024) would be crucial in radiation resistance as they would source non-enzymatic metabolites that help in ROS quenching.

- Presence of three homologs of *spx* regulatory genes in VITHBRA001 (against two homologs in VITHBRA024) a condition observed in *Priestia megaterium* (formerly known as *Bacillus megaterium*) which is reported to survive >5 kGy of gamma radiation [163, 182].

- VITHBRA001 has an operon dedicated to carotenoid pigment production which are efficient antioxidants; VITHBRA024, however, does not possess such genes.

- The COG and KEGG analyses reveal the existence of higher number of functional genes in VITHBRA001, in categories which are reported to show better radiation resistance, as mentioned in "Annotation of assembled genomes" section.

## Evolutionary perspective

The evolution of radio-resistance has been a subject of discussion for decades. From eco-evolutionary perspective, radio-resistance is thought to be an aftermath of resistance to several common stressors [183]. Radiation-resistant extremophiles are scattered and have been isolated from diverse environments [31], which indicates that the radiation-resistant extremophiles do not have a common ancestor. Deserts have been observed to be the most common habitat to find radiation-resistant extremophiles [184, 185]. There are suggestions on ROS-scavenging system as the interlinkage that enables the microbes to resist many different stresses including radiation [186]. There exists a general argument that radiation resistance has been attained through desiccation [187]. Apparently, this is applicable for most of the highly resistant varieties (including *Deinococcus sp*.). However, that desiccation as a stressor is applicable in the resistance of the candidate strains, is enigmatic, as these species have been exposed to low-dose radiation from time immemorial. Earlier, as a first-time study, our laboratory had isolated 35 bacterial species from this HBRA soil [32]. Out of these, 32, selected at random, were subjected to gamma irradiation to the tune of 1–5 kGy. Interestingly, eight of them had D10 value of >1 kGy and survived 5 kGy, indicating that a sizeable percentage (25%) of the bacterial population of this HBRA could be considered as IRRB. It is tempting to suggest that this sizeable proportion of radiation resistance, shown by the microbial population, could be the (hormetic) aftermath of long-term exposure to low-dose radiation prevalent in the HBRA.

In both the candidate species, we could observe instances of gene redundancy as well as functional redundancy that invites our attention, a matter of evolutionary significance. Gene level redundancy has been suggested to result in increased genetic robustness as it may 'protect' the organism in question from potentially harmful mutations, and help better environmental adaptation [188, 189]. It has already been proven that redundancies among genes involved in reactive oxygen species clearance can promote radio-resistance of *D. radiodurans*, inasmuch as deficient genes (mutant ones for example) could be substituted by a group of normal genes which are kept in reserve to act as soon as the situation demands [100]. One of the examples to be cited from the candidate specimen (VITHBRA001) for gene redundancy is UvrA (A1 and A2), involved in DNA repair; UvrA2 could also be involved in repair in the absence of UvrA1 [92]. Further, MnKat and catalases, involved in removal of reactive oxygen, and AddAB and RecQ RecJ, involved in DNA end-resection could also be listed under functional redundancy in the candidate species.

We also examined the possible role of pleiotropic genes, if any, in the candidate species since such genes are known to contribute to radiation resistance. A pleiotropic protein (PprA) from *D. radiodurans* has been found to substantially help in radiation resistance through DSB repair. Also, this gene when transfected in *E. coli* shows increased catalase activity assisting higher tolerance to $H_2O_2$ [190, 191]. Another pleotropic protein RecQ, with its three tandem HRDC domains, known to regulate DNA replication, recombination repair, nonhomologous end-joining, telomere maintenance, and damage response has been demonstrated to involve in radiation resistance [192]. Significantly, such pleiotropic genes are not ubiquitous, but thus far reported only from *D. radiodurans*. However, in the present study, *pnp* gene, known for its pleiotropic function has been observed in both the candidate strains. It has 3'-5' exoribonuclease activity to remove damaged ribonucleotides [193], basal trimming activity at the DSBs of DNA for HR and NHEJ pathways [98]. *D. radiodurans* is reported to have *pnp* gene, important in removing 8-oxo-ribonucleotide, formed after IR exposure and may also interact with other DNA repair genes [194].

## Conclusion

This study brings forward two species from HBRA, whose radiation resistance has not been reported thus far, and add to the knowledge on radiation resistant capabilities of the phylum Firmicutes. Results of the present study involving gamma irradiation experiments and the whole genome sequencing, involve almost all the domains of radiation resistance strategies, such as DNA repair, ROS quenching and protein protection. These results have not only offered us clear indications on the genetic machinery of two radiation-resistant bacterial strains inhabiting the HBRA, but it also provides us with valuable indications on the possible strategies of these microbes to combat radiation stress through various lines of defence. Additionally, the comparative genomic studies involving the candidate strains (VITH-BRA001 and VITHBRA024), *Deinococcus* species (radiation resistant) and *E. coli* (radiation sensitive), offer us on the one hand, key information on the differences in their genetic machinery, which in turn would help us explain the higher tolerance level shown by VITH-BRA001. Nevertheless, the results of the present Whole Genome Analysis (also involving COG) cover only ~80% of the genomic information of the candidate strains (*M. halosaccharovorans* and *B. paralicheniformis*), leaving about 20% of the genomic data as unidentified, a limitation with respect to the depth of the analysis, warranting future research. That these unidentified genes, at least some of them, could encompass those related to radiation resistance, cannot be ignored. Future research involving transcriptomic analysis should help us fill the gap substantially.

## Supporting information

**S1 Fig. Combined phylogenomics tree of two candidate strains.** The tree was built using TYGS platform using the closely allied type species of each strain (observed in Fig 2) and *D. radiodurans* as outgroup.
(TIF)

**S2 Fig. NCBI CD search analysis of VITHBRA001_02893.** Identifying the protein VITH-BRA001_02893 having MutS2 domain but the C-terminal region does not have the small MutS2 region (SMR) like the other homolog VITHBRA001_01182.
(TIF)

**S3 Fig. NCBI CD search analysis of MutS2 homolog.** VITHBRA001_01182 have the SMR region which can be seen in C-terminal region in this figure.
(TIF)

**S4 Fig. Multiple sequence alignment (MSA) of two homologs of MutS2.** The highlighted section (in yellow) shows the absence of small MutS2 region (SMR) in VITHBRA001_02893 as compared to VITHBRA001_01182. The MSA was performed using Clustal omega version 1.2.2.
(TIF)

**S5 Fig. Percentage Identity matrix of MutS2 protein for VITHBRA001 and VITHBRA024.** The MutS2 proteins for both the strains have been compared with MutS2 of *D. radiodurans* and *B. subtilis* to hypothecate the function of MutS2 of candidate strains. The aforementioned two species is reported to have different function of MutS2. The percentage identity suggests that the strains in this study have more identity with *B. subtilis* MutS2 as compared to *D. radiodurans*. Hence, the function of MutS2 of strains in this study could be similar to the function of *B. subtilis* MutS2. The deeper the color of the matrix the more the identity. The percentage identity was calculated using Uniprot align tool.
(TIF)

**S6 Fig. NCBI CD search analysis of VITHBRA001_00722.** It is identified as protein (VITH-BRA001_00722) belonging from the RuvC-like superfamily. This could be a part of RuvABC system that helps in resolution of Holliday junctions formed in homologous recombination repair of DNA.
(TIF)

**S7 Fig. Phylogenetic tree segregating AlkA and Mpg proteins.** Through this maximum likelihood phylogenetic tree, we could conclude that two proteins of VITHBRA001 belong to 3-methyladenine DNA glycosylase II (AlkA) whereas VITHBRA024 have one protein with 3-methyladenine DNA glycosylase I (Mpg) category and other with 3-methyladenine DNA glycosylase II (AlkA) category. The *D. radiodurans*'s proteins were used to identify the protein types. The tree was constructed using MEGA X with 1000 bootstrap support.
(TIF)

**S8 Fig. NCBI CD search analysis of bifunctional LigD.** Figures A and B belong to ligase LigD of VITHBRA001 and VITHBRA024 respectively. In both the cases we see that the proteins show presence of two function: the N-terminal ligase function and the C-terminal polymerase function.
(TIF)

**S9 Fig. NCBI CD search analysis of VITHBRA001_00018.** The analysis identifies the protein VITHBRA001_00018 as a MutT prototypical Nudix hydrolase protein which removes the

oxidized guanine product 8-oxoG form the free nucleotide pool.
(TIF)

**S10 Fig. Multiple sequence alignment of two homologs of AhpC observed in VITHBRA001 and VITHBRA024.** The first two sequences belong to atypical 3-cysteine containing AhpC and the last two belong to the typical 2-cysteine containing AhpC. The highlights (in yellow) show the presence of cysteine in the two types of AhpC. The MSA was performed using Clustal omega version 1.2.2.
(TIF)

**S11 Fig. Percentage identity matrix of Bcp proteins.** The three Bcp homologs in *D. radiodurans* are compared with the one Bcp protein identified in each VITHBRA001 and VITHBRA024. It was observed that the Bcp of the candidate strains have more identity with the *D. radiodurans*'s DR0846 Bcp protein. The deeper the colour in the matrix the more is the identity of the proteins. The percentage identity was calculated using Uniprot align tool.
(TIF)

**S12 Fig. Percentage identity matrix of BshB homologs of VITHBRA001 and VITHBRA024.** The matrix helped segregate the annotation of the two homologs of BshB genes present in the candidate strains with the help of BshB homologs of *B. subtilis* which has already been studied and reported. The deeper the colour in the matrix the more is the identity of the proteins. The percentage identity was calculated using Uniprot align tool.
(TIF)

**S13 Fig. NCBI CD search analysis of VITHBRA001_04484.** The sequence VITHBRA001_04484 is annotated as a homolog of thioredoxin reductase (TrxB) by Prokka which is identified and confirmed by NCBI CD search as Bacillithiol disulphide reductase (YpdA).
(TIF)

**S14 Fig. Trx-like proteins present in both VITHBRA001 and VITHBRA024.** Multiple sequence alignment shows the WCPDC motif (highlighted in yellow) present in the two *trx*-like genes of both the strains which is also the motif present in a Trx-like protein of *D. radiodurans* which is reported to be expressed after radiation stress.
(TIF)

**S15 Fig. Percentage identity of TrxB proteins of VITHBRA001 and VITHBRA024.** Three homologs of TrxB genes are present in VITHBRA001 and a comparison with *D. radiodurans* and *E. coli* TrxR gene helped to identify which homolog has more identity with TrxR of *D. radiodurans*. The deeper the colour in the matrix the more is the identity of the proteins. The percentage identity was calculated using Uniprot align tool.
(TIF)

**S16 Fig. Percentage identity matrix to differentiate the annotation of VITHBRA024 MntB proteins.** Through percentage identity it was observed that MntB1 had more identity to MntD of VITHBRA001 than to MntC. However, both the proteins (MntC and MntD) seem to have the permease activity in the MntABC complex hence, MntB1 would have permease activity. The deeper the colour in the matrix the more is the identity of the proteins. The percentage identity was calculated using Uniprot align tool.
(TIF)

**S1 File. Annotation information of curated genes that are responsible for radiation resistance in the organisms of this study.** The information is provided in an excel file having separate worksheets for both the candidate strains. The worksheets are also divided based to their

major function of "DNA repair" and "ROS quenching and protein protection". It also mentions additional genes which are responsible for replication and may give subsequent support during repair but they are not discussed in the text.
(XLSX)

**S1 Table. Standalone BLAST results between $C_{30}$ carotenoid biosynthesis proteins of various bacteria and protein FASTA file of VITHBRA001.**
(DOCX)

**S2 Table. Other repair, recombination and replication genes for candidate strains, *D. radiodurans* and *E. coli*.**
(DOCX)

## Acknowledgments

The authors would like to acknowledge Dr. S. Kartikeyan, Associate Professor, VIT, Vellore, for having shared the "Qiagen AllPrep® Bacterial DNA/RNA/Protein kit" for the extraction of bacterial DNA used in WGS analysis.

## Author Contributions

**Conceptualization:** Sowptika Pal, Anilkumar Gopinathan.

**Data curation:** Sowptika Pal.

**Formal analysis:** Sowptika Pal.

**Funding acquisition:** Jayanthi Abraham, Anilkumar Gopinathan.

**Investigation:** Sowptika Pal.

**Methodology:** Sowptika Pal, Ramani Yuvaraj, Hari Krishnan.

**Project administration:** Jayanthi Abraham, Anilkumar Gopinathan.

**Resources:** Balasubramanian Venkatraman, Jayanthi Abraham.

**Supervision:** Balasubramanian Venkatraman, Anilkumar Gopinathan.

**Validation:** Sowptika Pal.

**Visualization:** Sowptika Pal.

**Writing – original draft:** Sowptika Pal.

**Writing – review & editing:** Anilkumar Gopinathan.

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
