## [Decision Letter · Decision Letter 0]

13 Feb 2024

PONE-D-23-43346Unraveling radiation resistance strategies in two bacterial strains from the high background radiation area of Chavara-Neendakara: A comprehensive whole genome analysisPLOS ONE

Dear Dr. Gopinathan,

Thank you for submitting your manuscript to PLOS ONE. After careful consideration, we feel that it has merit but does not fully meet PLOS ONE’s publication criteria as it currently stands. Therefore, we invite you to submit a revised version of the manuscript that addresses the points raised during the review process. Manuscript has been reviewed by 2 subject experts. Both suggested major revision. Comments are appended below. You may find that they suggested some additional data. Authors are suggested to carefully address reviewers comments and submit revision for decision. Please submit your revised manuscript by Mar 29 2024 11:59PM. If you will need more time than this to complete your revisions, please reply to this message or contact the journal office at plosone@plos.org. Please include the following items when submitting your revised manuscript:A rebuttal letter that responds to each point raised by the academic editor and reviewer(s). You should upload this letter as a separate file labeled 'Response to Reviewers'.A marked-up copy of your manuscript that highlights changes made to the original version. You should upload this as a separate file labeled 'Revised Manuscript with Track Changes'.An unmarked version of your revised paper without tracked changes. You should upload this as a separate file labeled 'Manuscript'.

We look forward to receiving your revised manuscript.

Kind regards,

Hari S. Misra, Ph.D.

Academic Editor

PLOS ONE

Reviewers' comments:

Reviewer's Responses to Questions

**Comments to the Author**

1. Is the manuscript technically sound, and do the data support the conclusions?

Reviewer #1: Yes

Reviewer #2: Yes

2. Has the statistical analysis been performed appropriately and rigorously? 

Reviewer #1: No

Reviewer #2: No

3. Have the authors made all data underlying the findings in their manuscript fully available?

Reviewer #1: Yes

Reviewer #2: Yes

4. Is the manuscript presented in an intelligible fashion and written in standard English?

Reviewer #1: Yes

Reviewer #2: No

5. Review Comments to the Author

Reviewer #1: The manuscript by S. Pal and co-authors reports the genome analysis of two radiation resistant bacterial strains isolated from the high background radiation area of Chavara-Neendakara in India. The study presents the sequencing and assembly of the genomes of these two bacterial strains and an in-depth analysis of the factors that may contribute to their different levels of radiation resistance. The genomes are notably compared to that of the well-known radiation resistant bacteria Deinococcus radiodurans.

Overall, the work has been well performed and the analysis sheds light on the potential factors that may explain the differences in radiation resistance between the two strains. I, nonetheless, suggest a few changes and further analyses to strengthen and clarify the manuscript.

1.Introduction: (a) Remove the first paragraph that provides a short summary of the study, which is not needed. (b) page 3 line 62, ‘shown’ should be ‘known’. (c) page 3 line 64: DNA lesions are not caused by damaging nucleotide bases... this sentence needs to be rephrased. (d) page 4: it would be helpful to provide a clear description of the factors that have been identified in earlier work as being key determinants of radiation resistance and I would suggest referring to figure 6 at this point in the introduction (and therefore moving Fig. 6 to Fig. 1). (e) page 4 lines 94-97: this sentence is quite unclear. The authors should expand this sentence to better explain what they mean and the role of these different pathways for non-experts. Also explain what the differences are, in which organisms etc.

2.The word ‘discrepancies’ is used several times in this manuscript, but I think the authors mean ‘differences’. This should be corrected because the two words do not have the same meaning.

3.Acronyms should be explained and spelled out when first used.

4.Methods: what dose rate was used for the irradiation? How long did the irradiation last?

5.Results – Survival curves: The survival plots look somewhat strange. What is plotted? Are these the mean values of the triplicate measurements? The plot for VITHBRA024 is clearly not a linear regression; it actually seems to reach a plateau after a sudden drop. It is actually surprising to see a linear regression for the other strain, which is quite unusual. It would be good to have data for lower doses also.

6.Results – Fig. 2 & 3: figures 2 and 3 could be combined into a single figure with two panels. It would be interesting to know how distantly related the two strains are. Would it be possible to show a phylogenomic tree with both strains visible in addition to the more focused trees on each of the individual strains?

7.Results – Genome annotation: The authors used different annotation programs to analyse their assembled genomes. As a non-expert, which of these annotations can we trust? Can the authors briefly describe the differences? What was used for the rest of the study?

8.Results – Gene classification: As for figures 2 and 3, I suggest combining figures 4 and 5 into a single figure with 2 panels. It would be interesting and helpful for the comparison to include a radiation resistant (eg . Deinococcus radiodurans) and a radiation sensitive (eg. Escherchia coli) bacteria in these plots for comparison. To reliably compare the number of genes under different categories, the authors should use statistical tests and report on these to determine whether the observed differences are statistically significant or not.

9.Results – Genes involved in radiation resistance: (a) Remove first paragraph (lines 319-324 page 15), which is a repeat of the introduction, and simply introduce the analysis with a short sentence explaining the rationale for the analysis of selected radiation resistance factors. (b) What about the SOS response factors? The authors mention the alternative system found in Deinococcus (the RDRM system) and note that it is missing in their strains. So, do these two strains have a classical SOS response system? Also, there is no mention of the photolyase, an important enzyme for UV-induced DNA damage which is missing in D. radiodurans. (c) Importantly, this whole section would benefit from including D. radiodurans (as an example of a radiation resistant bacterium) and E. coli (as an example of a radiation sensitive bacterium) in the various tables presenting the genes potentially involved in radiation resistance (Tables 2 to 12). This would make it easier to compare and to highlight the genes that are associated with radiation resistance, those which are found in all bacteria and those which are missing in radiation resistant bacteria (eg RecBCD). Table 13 could then focus more particularly on the similarities and differences with D. radiodurans. (d) As it is written, it sounds like the presence of DNA repair genes is associated with radiation resistance, but this is incorrect and should be clarified. Most of the genes involved in DNA repair are present in all bacteria and are typically highly conserved. The authors should thus focus on the differences, eg. expanded families (UvrA2, extra UvrC, AlkA), high number of DNA glycosylases and Nudix hydrolases etc.

10.Conclusions: this could be shortened, the manuscript is already very long.

Reviewer #2: The manuscript titled "Unraveling radiation resistance strategies in two bacterial strains from the high background radiation area of Chavara-Neendakara: A comprehensive whole genome analysis" presents a detailed investigation into the genomic basis of radiation resistance in two bacterial strains, Metabacillus halosaccharovorans (VITHBRA001) and Bacillus paralicheniformis (VITHBRA024), isolated from a high background radiation area. The study is comprehensive, covering aspects from irradiation experiments to whole genome sequencing and analysis. However, there are several areas where the manuscript could be improved for clarity, scientific rigor, and overall contribution to the field:

1.The introduction lacks detailed discussion on studying mid-range radiation-resistant bacteria's significance, especially regarding biotechnological applications and environmental bioremediation. Elaborate on contextual importance and key knowledge gaps addressed.

2.Please include more information on statistical methods used for data analysis, including specific tests, confidence intervals, and significance levels related to survival curves, D10 values, and comparative genomics. This would strengthen reliability of conclusions drawn.

3.The section on comparison with Deinococcus radiodurans is useful. Please expand comparison analysis to include additional radiation-resistant and susceptible strains for broader evolutionary context elucidating genetic basis across bacterial lineages.

4.There is a lot of detail in the results section, and it makes it hard for the readers to comprehend the main points. Please consider moving some supplemental data to figures or tables and highlighting only the most significant findings in the main text.

5.Please ensure high quality, clear resolution for all figures and graphs. Carefully review and revise language for clarity and conciseness.

6.Experimental validation via gene knockouts or overexpression would significantly reinforce inferred roles of genes/pathways in radiation resistance.

7.Expand discussion of findings’ ecological and evolutionary implications, including potential evolution mechanisms of identified genes/pathways.

8.Expand discussion of findings’ ecological and evolutionary implications, including potential evolution mechanisms of identified genes/pathways. Address limitations in relying solely on D10 values as radiation resistance measure, depth of analysis, unidentified genes. Discuss future directions addressing these gaps.

9.The discussion of functional redundancy and pleiotropy of genes is lacking, making it difficult to definitively assign specific roles in radiation resistance.

Overall, the manuscript presents an interesting study on the genomic basis of radiation resistance in two bacterial strains from a high background radiation area. Addressing the above points would strengthen the manuscript, making it a valuable contribution to the field of microbial genomics and radiation biology.

6. PLOS authors have the option to publish the peer review history of their article (what does this mean?). If published, this will include your full peer review and any attached files.

Reviewer #1: No

Reviewer #2: No

---

## [Author Response · Author response to Decision Letter 0]

26 Apr 2024

The response to the comments from the Reviewers is being uploaded in the portal as "Response to Reviewers".

---

## [Editor Report · Decision Letter 1]

3 May 2024

PONE-D-23-43346R1Unraveling radiation resistance strategies in two bacterial strains from the high background radiation area of Chavara-Neendakara: A comprehensive whole genome analysisPLOS ONE

Dear Dr. Gopinathan,

Thank you for submitting your manuscript to PLOS ONE. After careful consideration, we feel that it has merit but does not fully meet PLOS ONE’s publication criteria as it currently stands. Therefore, we invite you to submit a revised version of the manuscript that addresses the points raised during the review process.

The revised manuscript has been checked for authors response to reviewers' comments, which have been largely addressed. Manuscript is much improved. There are some minor concerns that need to be addressed before final acceptance of the manuscript.

1. Attention is invited for rebuttal in response to Reviewer 1's comment 1e. How RecBCD presence or absence can influence recombination repair is well taken with examples. Does RecBC really has a role in DSB repair (believed in some bacteria) was addressed in D. radiodurans using E coli RecBC. A mentioned on this aspect may help to advance our understanding on the paradigm that RecBC(D) as a recombination repair enzyme. 

2. Rev.1, comment 4. Dose rate in this work is found to be moderate (.77kGy/h). Data compared from others are mostly high dose rate. While some work has also been done in D. radiodurans with low dose rate. Authors should try to discuss dose rate factor with appropriate citation.

3. Some of the rebuttal components are not seen in revised tracked copy of the manuscript. May recheck it please. 

4. Although, PLOS ONE does not restrict to the size of the paper and references, the quality of referencing is advised. I could see some of the citations are redundant and can be removed. Please check the quality of work, journals and originality in the work that are being cited in manuscript. If possible, bring down number of citations to close to 150 from the current number more than 200.

6. I suggest checking the manuscript carefully for English language and composition.  

We look forward to receiving your revised manuscript.

Kind regards,

Hari S. Misra, Ph.D.

Academic Editor

PLOS ONE

Journal Requirements:

Additional Editor Comments:

The revised manuscript has been checked for authors response to reviewers' comments, which have been largely addressed. Manuscript is much improved. There are some minor concerns that need to be addressed before final acceptance of the manuscript.

1. Attention is invited for rebuttal in response to Reviewer 1's comment 1e. How RecBCD presence or absence can influence recombination repair is well taken with examples. Does RecBC really has a role in DSB repair (believed in some bacteria) was addressed in D. radiodurans using E coli RecBC. A mentioned on this aspect may help to advance our understanding on the paradigm that RecBC(D) as a recombination repair enzyme.

2. Rev.1, comment 4. Dose rate in this work is found to be moderate (.77kGy/h). Data compared from others are mostly high dose rate. While some work has also been done in D. radiodurans with low dose rate. Authors should try to discuss dose rate factor with appropriate citation.

3. Some of the rebuttal components are not seen in revised tracked copy of the manuscript. May recheck it please.

4. Although, PLOS ONE does not restrict to the size of the paper and references, the quality of referencing is advised. I could see some of the citations are redundant and can be removed. Please check the quality of work, journals and originality in the work that are being cited in manuscript. If possible, bring down number of citations to close to 150 from the current number more than 200.

6. I suggest checking the manuscript carefully for English language and composition.

---

## [Author Response · Author response to Decision Letter 1]

11 May 2024

Specific answers to all the queries of Reviewer/Editor are uploaded in the editorial manager with the file name "Response to Reviewers".

---

## [Editor Report · Decision Letter 2]

16 May 2024

PONE-D-23-43346R2Unraveling radiation resistance strategies in two bacterial strains from the high background radiation area of Chavara-Neendakara: A comprehensive whole genome analysisPLOS ONE

Dear Dr. Gopinathan,

Thank you for submitting your manuscript to PLOS ONE. After careful consideration, we feel that it has merit but does not fully meet PLOS ONE’s publication criteria as it currently stands. Therefore, we invite you to submit a revised version of the manuscript that addresses the points raised during the review process. Please submit your revised manuscript by Jun 30 2024 11:59PM. If you will need more time than this to complete your revisions, please reply to this message or contact the journal office at plosone@plos.org. Please include the following items when submitting your revised manuscript:A rebuttal letter that responds to each point raised by the academic editor and reviewer(s). You should upload this letter as a separate file labeled 'Response to Reviewers'.A marked-up copy of your manuscript that highlights changes made to the original version. You should upload this as a separate file labeled 'Revised Manuscript with Track Changes'.An unmarked version of your revised paper without tracked changes. You should upload this as a separate file labeled 'Manuscript'.If applicable, we recommend that you deposit your laboratory protocols in protocols.io to enhance the reproducibility of your results. Protocols.io assigns your protocol its own identifier (DOI) so that it can be cited independently in the future. For instructions see: https://journals.plos.org/plosone/s/submission-guidelines#loc-laboratory-protocols. Additionally, PLOS ONE offers an option for publishing peer-reviewed Lab Protocol articles, which describe protocols hosted on protocols.io. Read more information on sharing protocols at https://plos.org/protocols?utm_medium=editorial-email&utm_source=authorletters&utm_campaign=protocols.

We look forward to receiving your revised manuscript.

Kind regards,

Hari S. Misra, Ph.D.

Academic Editor

PLOS ONE

Journal Requirements:

Additional Editor Comments:

Please check details and authenticity of information given in Table 13. Some exclusive genes in your isolates seems to be there in Deinoocccus radiodurans (e.g. UvsE, J Bacteriol. 2002 Feb; 184(4): 1003–1009).

---

## [Author Response · Author response to Decision Letter 2]

17 May 2024

The comments from the Reviewer have been addressed and the manuscript has been revised accordingly. The response to Reviewers has been uploaded in the editorial manager.

---

## [Editor Report · Decision Letter 3]

20 May 2024

Unraveling radiation resistance strategies in two bacterial strains from the high background radiation area of Chavara-Neendakara: A comprehensive whole genome analysis

PONE-D-23-43346R3

Dear Dr. Gopinathan,

We’re pleased to inform you that your manuscript has been judged scientifically suitable for publication and will be formally accepted for publication once it meets all outstanding technical requirements.

Kind regards,

Hari S. Misra, Ph.D.

Academic Editor

PLOS ONE
---

## [Editor Report · Acceptance letter]

30 May 2024

PONE-D-23-43346R3 

PLOS ONE

Dear Dr. Gopinathan, 

I'm pleased to inform you that your manuscript has been deemed suitable for publication in PLOS ONE. Congratulations! Your manuscript is now being handed over to our production team.

Kind regards, 

on behalf of

Professor Hari S. Misra 

Academic Editor

PLOS ONE